# Image and Video Quality Assessment using Prompt-Guided Latent Diffusion Models for Cross-Dataset Generalization

**Shankhanil Mitra**                                                              *sk.mitra@samsung.com*
*Samsung Research Institute, Bangalore*

**Diptanu De**                                                                   *diptde@qti.qualcomm.com*
*Qualcomm, Hyderabad*

**Shika Rao**                                                                    *sr7463@nyu.edu*
*Courant Institute of Mathematical Science*
*New York University*

**Rajiv Soundararajan**                                                          *rajivs@iisc.ac.in*
*Department of Electrical Communication Engineering*
*Indian Institute of Science*

**Reviewed on OpenReview:** *https://openreview.net/forum?id=FjhvVevAoQ*

## Abstract

The design of image and video quality assessment (QA) algorithms is extremely important to benchmark and calibrate user experience in modern visual systems. A major drawback of the state-of-the-art QA methods is their limited ability to generalize across diverse image and video datasets with reasonable distribution shifts. In this work, we leverage the denoising process of diffusion models for generalized image QA (IQA) and video QA (VQA) by understanding the degree of alignment between learnable quality-aware text prompts and images or video frames. In particular, we learn cross-attention maps from intermediate layers of the denoiser of latent diffusion models (LDMs) to capture quality-aware representations of images or video frames. Since applying text-to-image LDMs for every video frame is computationally expensive for videos, we only estimate the quality of a frame-rate sub-sampled version of the original video. To compensate for the loss in motion information due to frame-rate sub-sampling, we propose a novel temporal quality modulator. Our extensive cross-database experiments across various user-generated, synthetic, low-light, frame-rate variation, ultra high definition, and streaming content-based databases show that our model can achieve superior generalization in both IQA and VQA.

## 1 Introduction

The proliferation of mobile devices with image and video capturing capabilities has led to an explosion in the number of images and videos captured, stored and shared on various platforms. This has necessitated quality assessment (QA) of images and videos. QA algorithms are broadly classified into three major categories, namely full-reference QA, reduced-reference QA and no-reference QA. 1) Full-reference QA: Here, the pristine/undistorted version of the inference image/video is also available. Examples of such measures include SSIM Wang et al. (2004), and VMAF Rassool (2017). 2) Reduced-reference QA: Here, only some partial information about the pristine version of the inference image/video is also available. RRED Soundararajan & Bovik (2011) and ST-RRED Soundararajan & Bovik (2013) are examples of such methods. 3) No-reference

---

§https://github.com/Shankhanil006/GenzIVQA
*Work done at Indian Institute of Science.

QA: Here, only the inference image/video is available for quality assessment. NR QA has applications in assessing the quality of images/videos that are captured using various camera devices, where a pristine version of that image/video is absent.

Several classical NR algorithms for image QA (IQA) (Mittal et al., 2012) and video QA (VQA) (Saad et al., 2014), suffer in their ability to model a wide range of distortions. The emergence of deep neural networks (DNN) gave rise to a variety of NR-IQA (Ke et al., 2021; Su et al., 2020) and NR-VQA (Li et al., 2019a; 2021) methods. The DNN based methods suffer from a lack of generalization capability. Such models trained on a large dataset fail to predict image or video quality on other datasets accurately. For example, models trained on a large dataset with camera captured videos fail to generalize to varying evaluation scenarios such as diverse camera captures, varying frame rates, gaming videos, ultra high-definition and so on. Multimodal vision-language models were recently shown to be promising for their generalizability for NR-IQA and NR-VQA. In particular, CLIP-IQA (Wang et al., 2023a) and BUONA-VISTA (Wu et al., 2023a) show the capacity of vision-language models to predict image and video quality respectively even in a zero-shot setting. Such models can achieve promising cross-database generalizability on par with IQA and VQA specific models through a cost-effective prompt tuning method. These observations motivate the study of how to leverage existing large pretrained models to achieve cross-database generalizable NR-QA.

Recently, several pieces of work find that text-to-image (T2I) diffusion models show superior out of distribution generalization performance compared to vision language models on a variety of image retrieval, recognition, and reasoning tasks (He et al., 2023; Li et al., 2023a; Kawar et al., 2023; Ma et al., 2023). This makes them an interesting choice for achieving generalizable NR-QA. The reason for such generalization has been attributed to the inductive bias in the denoising architecture (Kadkhodaie et al., 2024). However, diffusion models are typically designed for generation of different image content conditioned on textual descriptions and their application to the QA task is non-trivial. In particular, arbitrary noise levels cannot help extract quality aware features. Moreover, since diffusion models are trained to exploit the textual prompts for generation, there is a need to understand how textual prompts can be used to extract quality-aware information.

In this work, we present Generalized IQA (**GenzIQA**) and Generalized VQA (**GenzVQA**) to explore the potential of prompt-guided T2I latent diffusion models (LDMs) for achieving cross-database generalizability in both NR-IQA and NR-VQA respectively. In particular, we leverage the generalization capability of diffusion models for the IQA task by using cross-attention finetuning, quality-aware prompt learning, and the design of suitable noise levels to extract quality-aware features from intermediate layers of the denoiser of the reverse diffusion process. We conduct a detailed analysis to show that there is a delicate choice of the noise to be added to the image when passed through the diffusion model to extract quality-aware features. Thus, our key contribution is to effectively leverage the generalization capabilities of LDMs for the IQA task. We show that our approach can achieve far superior cross-database generalization than any existing NR-IQA model on a variety of datasets. Applying such a T2I model to every video frame for VQA is computationally expensive. In this regard, we estimate the quality of the video at a lower frame-rate but compensate for the loss in motion information in the sub-sampled video. In particular, we propose a temporal quality modulator (TQM) that adjusts the predicted video quality by accounting for the loss of motion information. TQM estimates how the similarities of the visual features given by diffusion model with motion features of sub-sampled video differ from that of its similarities with the motion-features of original video. Our main contributions are summarized below:

- We design a unified framework for NR-IQA and NR-VQA to achieve the best cross-database generalizable performance among all existing methods across a variety of IQA and VQA datasets.

- We show that quality-aware tuning of cross-attention maps, extracted from the intermediate layers of the denoiser in the reverse diffusion process, in conjunction with quality-aware learning of contextual text prompts are necessary to render diffusion models effective for IQA and VQA.

- We propose a novel temporal quality modulator by computing the cross-attention between the sub-sampled video features of the LDM and the video motion features at original and sub-sampled frame rates. This allows the LDM to estimate video quality at reasonable compute times.

- We conduct a detailed analysis of the role of noise added to the latent variable during denoising and find that there exists a delicate relationship between the noise level and the ability of the denoiser for effective QA.

- We perform extensive experiments across 11 VQA and 6 IQA databases covering user-generated, restoration, variable frame-rate, Ultra-HD, and streaming video scenarios to establish the superior cross-database generalizability of our model with respect to existing models.

## 2 Related Work

### 2.1 Image Quality Assessment

Hand-crafted feature-based methods such as BRISQUE (Mittal et al., 2012), DIIVINE (Moorthy & Bovik, 2011) and BLIINDS (Saad et al., 2012), exploit the natural scene statistics while CORNIA (Ye et al., 2012) and HOSA (Xu et al., 2016) design codebook learning-based methods. With the emergence of DNN, various end-to-end learning methods (Zhang et al., 2018c; Kim & Lee, 2016), or methods regressing pretrained convolutional neural network features (Zhang et al., 2018b; Zeng et al., 2017) against quality have been designed. Transformer-based models such as MUSIQ (Ke et al., 2021) and TReS (Golestaneh et al., 2022) also show promising performance on both synthetic and in-the-wild IQA tasks. MetaIQA (Zhu et al., 2020) proposes meta-learning for complex real-world distortions while HyperIQA (Su et al., 2020) proposes a hyper network to capture various distortion and semantic attributes in images. Recently, a few works employ diffusion models (Fu et al., 2024; Li et al., 2024b), but they involve training the entire diffusion model, thus increasing the computational complexity.

One approach to deal with generalization in IQA is by designing self-supervised quality representations through models such as CONTRIQUE (Madhusudana et al., 2022b), Re-IQA (Saha et al., 2023), and QPT (Zhao et al., 2023). CLIP-IQA (Wang et al., 2023a) is a vision-language model that shows very good zero-shot generalization for the IQA task. DEIQT (Qin et al., 2023) designs an attention-panel decoder learning with limited data samples. LIQE (Zhang et al., 2023a) trains a CLIP-based vision language model on six different databases, showing good performance in cross-database settings. TTA-IQA (Roy et al., 2023) uses the test-time adaptation technique to generalize a pretrained IQA model for different kinds of databases. Recently, GRepQ (Srinath et al., 2024) presents a self-supervised learning method that can lead to generalized quality representations. QCN (Shin et al., 2024) proposes a geometric order learning to achieve good cross-database performance in IQA. LoDa (Xu et al., 2024) adapts vision-transformers for IQA using another pre-trained CNN, while DSMix (Shi et al., 2024) proposes distortion-induced pre-training to enhance performance for existing IQA models. Recently, DiffV2IQA (Wang et al., 2025) proposed a dual branch model consisting of vision-transformer and ResNet50 to illustrate the correlation between diffusion model's ability to reconstruct an image and its quality. Also, PFD-IQA (Li et al., 2025) proposes IQA method by leveraging the denoising ability of a diffusion model to remove noise from quality-aware features. Despite these efforts, there is a need to consistently achieve better generalization across diverse and complex distortion types.

### 2.2 Video Quality Assessment

Classical approaches such as VBLIINDS (Saad et al., 2014) and VCORNIA (Xu et al., 2014) learn natural scene statistics of videos by modelling the discrete cosine transform. TLVQM (Korhonen, 2019) shows robust VQA performance by modelling temporal low complexity features with spatial high complexity features. VIDEVAL (Tu et al., 2021a) is an ensemble of various handcrafted features designed to capture diverse quality attributes in a video. Among DNN approaches, while VSFA (Li et al., 2019a) and MDTVSFA (Li et al., 2021) learn a gated recurrent unit on top of pretrained ResNet50 features (He et al., 2016), PVQ (Ying et al., 2021) learns an ensemble of ResNet50 trained on IQA and a 3D ResNet18 trained on action recognition tasks. CSVT-BVQA (Li et al., 2022) also transfers spatial knowledge from a pretrained IQA model and temporal knowledge from a pretrained action recognition model. Among transformer based models, FAST-VQA (Wu et al., 2022) learns an end-to-end model by spatially fragmenting the video clips which is extended to DOVER (Wu et al., 2023b) by incorporating aesthetics. SSL-VQA (Mitra & Soundararajan, 2024) learns a similar end-to-end model with limited labelled videos. KSVQE (Lu et al., 2024) employs CLIP (Radford

et al., 2021) in its design while ModularVQA (Wen et al., 2024) also uses a CLIP model along with a spatial and temporal quality rectifier. However all these methods do not generalize well across diverse distortions.

To address generalizability, VISION (Mitra & Soundararajan, 2022), and CONVIQT (Madhusudana et al., 2022a) present self-supervised learning based quality-aware feature extractors. VQA methods such as STEM (Kancharla & Channappayya, 2022), VIQE (Zheng et al., 2022), VISION (Mitra & Soundararajan, 2022), and TPQI (Liao et al., 2022) do not require any human labelled videos in their design and give reasonable quality estimates for user-generated content (UGC) videos. Nevertheless, their performance with respect to the methods trained with human opinion scores is sub-par.

## 3 Quality Assessment using Latent Diffusion Models

We first discuss the preliminaries of latent diffusion models, followed by the procedure on how to adapt such models for IQA and VQA.

### 3.1 Preliminaries of Latent Diffusion Models

Latent diffusion models (LDMs) (Rombach et al., 2022) are a class of diffusion models that encode a real image $x$ onto a low-dimensional latent space $z$ and learn a distribution in the latent space conditioned on a text input $y$. In particular, the forward process starts at an image latent variable $z_0$ progressively corrupted by Gaussian noise, and a learned reverse process generates samples from the latent distribution using a denoising model conditioned on $y$. In LDMs, the image $x$ is encoded as $z_0 = \varepsilon(x)$, where $\varepsilon(\cdot)$ is a vector quantized variational autoencoder (VQ-VAE). The generated latent samples are then passed through a decoder for image generation. Given any timestep $t$, the forward process distorts the latent representation $z_0$ to a noisy latent $z_t$ as

$$z_t = \sqrt{\bar{\alpha}_t} z_0 + \sqrt{(1 - \bar{\alpha}_t)} \epsilon, \tag{1}$$

where $\epsilon \sim \mathcal{N}(0, \mathbf{I})$, $\bar{\alpha}_t = \prod_{s=1}^{t} \alpha_s$, $\alpha_t = 1 - \beta_t$ and $\{\beta\}_{t=1}^{T}$ are the noise variances at every timestep $t \in \{1, 2, \cdots, T\}$ in the forward process. In the reverse process, the denoising autoencoder $\epsilon_\theta(\cdot)$ takes in the noisy latent $z_t$, timestep variable $t$ and the conditional variable $y$ to estimate the additive noise in the forward process.

Given a text prompt $y$, let the CLIP text encoder output be $\tau_\theta(y) \in \mathbb{R}^{M \times d_\tau}$, where $M$ is the number of text tokens and $d_\tau$ is the feature dimension. For a noisy latent $z_t$, let $\varphi_p(z_t) \in \mathbb{R}^{N^p \times d_\epsilon^p}$ be the intermediate (flattened) visual representation at block $p$, $p \in \{1, 2, \cdots, L\}$, in the denoiser UNet $\epsilon_\theta(\cdot)$, where $N^p$ is the number of visual tokens. The intermediate cross-attention block of UNet maps the text representation onto the image representation for feature generation through the operation

$$\text{Attention}(Q^{(p)}, K^{(p)}, V^{(p)}) = \text{softmax}\left(\frac{Q^{(p)} K^{(p)^T}}{\sqrt{d}}\right) V^{(p)},$$

where $Q^{(p)} = W_Q^{(p)} \cdot \varphi_p(z_t)$, $K^{(p)} = W_K^{(p)} \cdot \tau_\theta(y)$ and $V^{(p)} = W_V^{(p)} \cdot \tau_\theta(y)$ are the query, key and value matrices with $W_Q^{(p)} \in \mathbb{R}^{d \times d_\epsilon^p}$, $W_K^{(p)} \in \mathbb{R}^{d \times d_\tau}$ and $W_V^{(p)} \in \mathbb{R}^{d \times d_\tau}$ being the respective projection matrices. $d$ is a hyper-parameter corresponding to the number of channels in each head of the multi-head cross-attention. Note that, the dot operation shown above in the expression for $Q^{(p)}, K^{(p)}$, and $V^{(p)}$ is a linear operation and can be expanded as $Q^{(p)} = W_Q^{(p)} \cdot \varphi_p(z_t) = \varphi_p(z_t)(W_Q^{(p)})^T$ and similarly for $K^{(p)}$ and $V^{(p)}$.

### 3.2 Image Quality Assessment using LDM

Stable Diffusion model (SDM) Rombach et al. (2022) is trained with image-text pairs from LAION-5B Schuhmann et al. (2022) dataset. Such datasets include perceptual attributes in its prompts such as detailed descriptions that are related to visual quality, and aesthetic-style captioning. Moreover, the 5 billion dataset size encompasses diverse content and quality. While generating high-quality images from noisy latent features using text-guided prompts, the generative pipeline progressively retrieves perceptual attributes at various stages of the UNet. In this work, we seek to extract such intermediate generalized visual representations

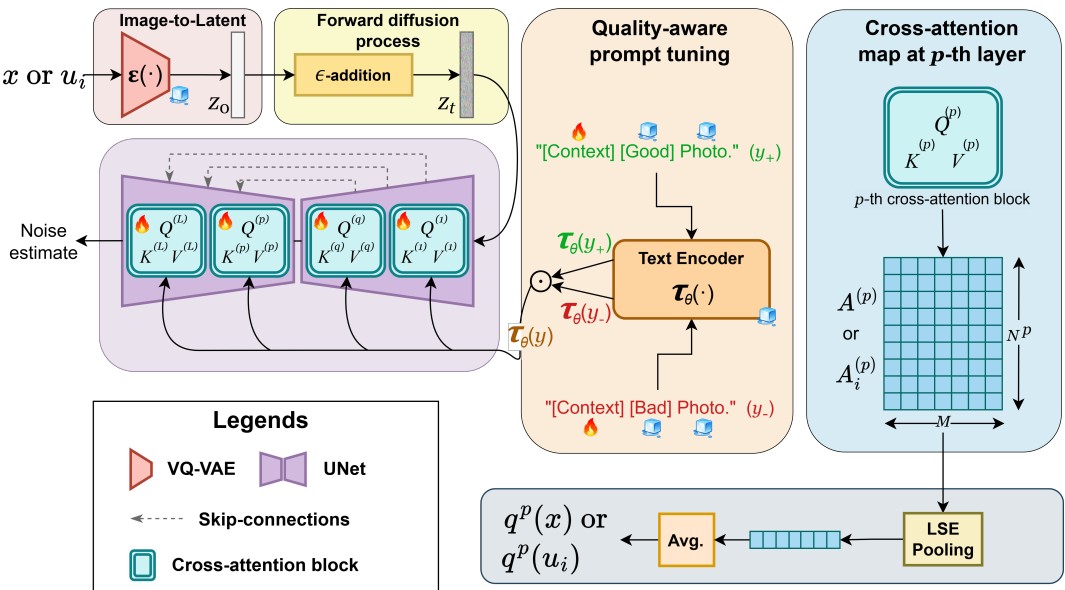

Figure 1: Given an input image $x$ or video frame $u_i$, VQ-VAE processes it to the latent $z_0$. The noisy latent output $z_t$ of the forward diffusion is fed to the denoising UNet (Ronneberger et al., 2015) $\epsilon_\theta(\cdot)$. At every cross-attention block in $\epsilon_\theta(\cdot)$, the intermediate visual representation is aligned with learnable text representations $\{\tau_\theta(y_+), \tau_\theta(y_-)\}$. After that, the attention maps are pooled for each cross-attention block $p$ to predict block quality $q^p(x)$ or $q^p(u_i)$.

from UNet for the quality assessment task. To specifically retrieve quality-aware features, we rely upon using cross-attention block outputs, since textual representations can be leveraged to capture quality-specific information from the UNet. As argued by Kumari et. al. Kumari et al. (2023) , cross-attention parameters are more sensitive to finetuning of diffusion models over self-attention and other parameter weights.

Thus, our GenzIQA model exploits the generalization capabilities of the LDMs (Rombach et al., 2022) for IQA by learning cross-attention maps between the image and text features in the reverse diffusion process in conjunction with quality-aware prompts to match the visual concepts. In particular, we tap into the reverse diffusion process, where the UNet denoises noisy features as shown in Fig. 1. We train all the cross-attention modules in the denoising UNet of the LDM to output quality estimates at different scales. We use the LDM to obtain quality $q(x)$ for image $x$. Given the visual representation of image $x$ at block $p$ as $\varphi_p(z_t)$ and textual representation as $\tau_\theta(y)$, we compute the attention map at every block $p$ of the UNet encoder and decoder as

$$A^{(p)} = \text{softmax}\left(\frac{Q^{(p)}K^{(p)^T}}{\sqrt{d}}\right), \tag{2}$$

where $A^{(p)} \in \mathbb{R}^{N^p \times M}$ measures the similarity between the visual query at the $p^{th}$ block of the UNet and the textual embedding of the text encoder. In our framework, we only learn the cross-attention weights.

We estimate the image quality by applying a log-sum-exponential (LSE) pooling of the attention map at every scale leveraging upon its robustness benefits (He et al., 2023). The predicted quality of $x$ at cross-attention block $p$ is given as

$$q^p(x) = \frac{1}{M}\sum_{m=1}^{M}\frac{1}{\lambda}\log\left(\sum_{n=1}^{N^p}\exp(\lambda A^{(p)}_{n,m})\right), \tag{3}$$

where $A_{n,m}^{(p)}$ is the attention score at location $(n, m)$. $\lambda$ is a temperature co-efficient used to amplify the similarity measure between the image and text features. The overall quality of image $x$ is estimated as

$$q(x) = \frac{1}{L} \sum_{p=1}^{L} q^p(x). \tag{4}$$

We obtain $q(x)$ through a **single-step denoising** of $z_t$. The noise added to the latent features $z_0$, specified through $t$, can have a significant impact on the ability of the LDM to predict image quality. There is a delicate relationship between the denoising ability of the UNet and the amount of additive noise, which could alter the semantic information and image quality information in the latent image representation. We investigate this relationship in our experiments later.

We jointly optimize the projection matrices of the UNet $\epsilon_\theta$, and the prompt embedding layer of $\tau_\theta$ using a mean squared error (MSE) loss between the ground-truth mean-opinion score (MOS), available in the training dataset and $q(x)$ as

$$\mathcal{L}_{\mathcal{IQA}} = ||q(x) - \text{MOS}(x)||_2^2. \tag{5}$$

**Contextual Prompt Tuning:** We further enhance the ability of the cross-attention maps to model quality through prompt-tuning. Prompt-tuning not only saves computational resources but also preserves the generalization capability of the text encoder. Similar to CLIP-IQA$^+$, we design a pair of context prompts with 'Good Photo' and 'Bad Photo' as the initial positive and negative attributes. Our final learnable context is given as

$$y_+ = [Context] + \text{Positive Attribute} \quad \text{and}$$
$$y_- = [Context] + \text{Negative Attribute}, \tag{6}$$

where the $[Context]$ is a sequence of 16 tokens learned using CoOp (Zhou et al., 2022). Thus the text input $y$ chosen for the IQA task is given by the pair $\{y_+, y_-\}$. We take the **average** of the two quality predictions corresponding to $\{y_+, y_-\}$, to estimate the final quality of the given image as we view the two quality predictions as estimates from two diverse quality-aware text prompts. Our initial (positive and negative) attributes give faster training convergence and also more accurate results rather than using only learnable contexts. While using a single prompt for representing quality, say 'Good Photo.', images with bad quality can manifest in various ways such as blurriness or compression or ringing artifacts. Such diverse notions of bad quality can make it challenging to align bad quality features together. Thus, specifying bad quality using an initial context such as 'Bad Photo.', helps to align all these different forms of highly distorted images into a unified representation. The same argument also applies to good quality features if only 'Bad Photo.' is used as a single prompt. Note that no image/video specific textual descriptions are used either during training or inference, only a generic textual attribute pair is used for capturing perceptual representations.

### 3.3 Video Quality Assessment using LDM

Applying GenzIQA (using Stable Diffusion Model v2) on every video frame can take several minutes to process a 10-second long 1080p video clip on a NVIDIA RTX 3090 GPU. To alleviate the long inference time with respect to video duration, we estimate the video quality at 1 frame per second (fps) and then compensate for this subsampling.

**Learning Sparse Video Quality:** Consider a video clip $v = \{\bar{u}_j\}_{j=1}^{T_o}$, where $\bar{u}_j \in \mathbb{R}^{H \times W \times 3}$ is the $j^{th}$ frame of the video $v$ and $T_o$ is the total number of frames. We sub-sample the video $v$ at one fps to get a sparse video clip $v_s = \{u_i\}_{i=1}^{T_s}$, where $T_s$ is the number of frames of the sub-sampled video $v_s$ and $u_i$ is the $i^{th}$ sub-sampled frame. Given the visual representation of frame $u_i$ at block $p$ as $\varphi_p(z_t^i)$, we compute the attention map at every block $p$ similar to GenzIQA as $A_i^{(p)} = \text{softmax}\left(\frac{Q_i^{(p)} K_i^{(p)T}}{\sqrt{d}}\right)$. The predicted quality of $v_s$ at cross-attention block $p$ is given as $q^p(v_s) = \frac{1}{T_s} \sum_{i=1}^{T_s} q^p(u_i)$, where

$$q^p(u_i) = \frac{1}{M} \sum_{m=1}^{M} \frac{1}{\lambda} \log \left( \sum_{n=1}^{N^p} \exp(\lambda A_{i,n,m}^{(p)}) \right), \tag{7}$$

and $A_{i,n,m}^{(p)}$ is the attention value at location $(n, m)$ for the frame $u_i$. The overall quality of the sub-sampled video $v_s$ is estimated as $q(v_s) = \frac{1}{L} \sum_{p=1}^{L} q^p(v_s)$.

**Temporal Quality Modulator (TQM):** Sub-sampling the video at one fps can lead to inaccurate estimation of the video quality (Wen et al., 2024). The temporal distortions pertaining to video motion cannot be accurately measured through a sparse sampling of the frames. Hence, we propose a TQM that fixes the quality estimate as shown in Fig. 2. Several recent works (Li et al., 2022; Sun et al., 2022; Wen et al., 2024) use the pre-trained SlowFast (Feichtenhofer et al., 2019) network to estimate motion-based quality in videos. The SlowFast network is a dual-stream model with a *Slow pathway*, operating at a low frame rate, and a *Fast pathway*, operating at a high frame rate. In our work, we utilize the slow and the fast pathways to estimate the video quality features at one fps and the actual frame rate, respectively. The SlowFast network is much smaller than the diffusion model and does not add much computational time.

We consider the SlowFast network that has been pre-trained on the video action recognition dataset viz. Kinetics-400K Kay et al. (2017). The slow pathway in SlowFast network is designed to capture spatial information in video frames from sub-sampled videos, while the fast pathway is designed to capture temporal information from the full-length video. As argued by Feichtenhofer et. al. Feichtenhofer et al. (2019), the fast pathway has better temporal modeling capability when compared to the slow pathway. Thus, we measure the similarity of the visual query across all frames of $v_s$ with the Slow and Fast pathway video features, respectively. We estimate how the similarities of the visual query across all frames of $v_s$ with the slow features differ with its similarities of the fast features by passing both similarity scores through a network to obtain a scalar quality modulation factor.

We extract the slow and fast pathway features from the pre-final layer of each pathway before each individual pathway's features are concatenated in the SlowFast network. Let $h_s(\cdot)$ and $h_f(\cdot)$ be the feature encoders for slow and fast pathways respectively. Then, the features corresponding to the sub-sampled and original videos are given as $h_s(v_s) \in \mathbb{R}^{T_s \times 256}$ and $h_f(v) \in \mathbb{R}^{T_o \times 2048}$. Let the query and key matrices at block $p$ corresponding to the Slow pathway be $Q_s^{(p)}$ and $K_s^{(p)}$. Further, let $W_{sQ}^{(p)}$ and $W_{sK}^{(p)}$ be the corresponding weight matrices. Also, define frame-level quality $R_i^{(p)} = W_{sQ}^{(p)} \cdot \varphi_p(z_t^i)$ for $i \in \{1, 2, \cdots, T_s\}$. Thus, $R_i^{(p)} \in \mathbb{R}^{N^p \times d}$. We spatially average pool the frame-level queries along the visual-token dimension to obtain $Q_s^{(p)}$ as

$$Q_s^{(p)}(i, m) = \frac{1}{N^p} \sum_{n=1}^{N^p} R_{i,n,m}^{(p)}, \tag{8}$$

where $m \in \{1, 2, \cdots, d\}$. Therefore, $Q_s^{(p)} \in \mathbb{R}^{T_s \times d}$. Note that $K_s^{(p)} = W_{sK}^{(p)} \cdot h_s(v_s)$ and hence $K_s^{(p)} \in \mathbb{R}^{T_s \times d}$. Similarly, we get the query and key matrices at block $p$ corresponding to the Fast features as $Q_f^{(p)} \in \mathbb{R}^{T_s \times d}$ and $K_f^{(p)} \in \mathbb{R}^{T_o \times d}$ with weight matrices $W_{fQ}^{(p)}$ and $W_{fK}^{(p)}$.

We get the attention maps between the denoising UNet and the SlowFast network features as $A_s^{(p)} = \text{softmax}\left(\frac{Q_s^{(p)} K_s^{(p)T}}{\sqrt{d}}\right)$ and $A_f^{(p)} = \text{softmax}\left(\frac{Q_f^{(p)} K_f^{(p)T}}{\sqrt{d}}\right)$, where $A_s^{(p)} \in \mathbb{R}^{T_s \times T_s}$ and $A_f^{(p)} \in \mathbb{R}^{T_s \times T_o}$ are the cross-attention maps with respect to the Slow and Fast pathway features respectively. We average pool the attention maps across spatial dimensions for all the cross-attention blocks $p \in \{1, 2, \ldots, L\}$ to get the cross-attention scores between the UNet feature and the SlowFast features as $S^p$ and $F^p$. We concatenate the scores $S^p$ and $F^p$ and pass them through a single-layer network $\phi^p$ to get a temporal quality correction factor as $\gamma^p = \phi^p(S^p, F^p)$. We estimate the quality of the original video as

$$q(v) = \frac{1}{L} \sum_{p=1}^{L} \gamma^p q^p(v_s). \tag{9}$$

We optimize the projection matrices of the UNet $\epsilon_\theta$, the prompt embedding layer of $\tau_\theta$, TQM's cross-attention matrix weights $(W_{sQ}^{(p)}, W_{sK}^{(p)}, W_{fQ}^{(p)}, W_{fK}^{(p)})$ and $\phi^p$ for all $p$ using MSE loss between the ground-truth MOS and $q(v)$ as

$$\mathcal{L}_{\mathcal{VQA}} = ||q(v) - \text{MOS}(v)||_2^2. \tag{10}$$

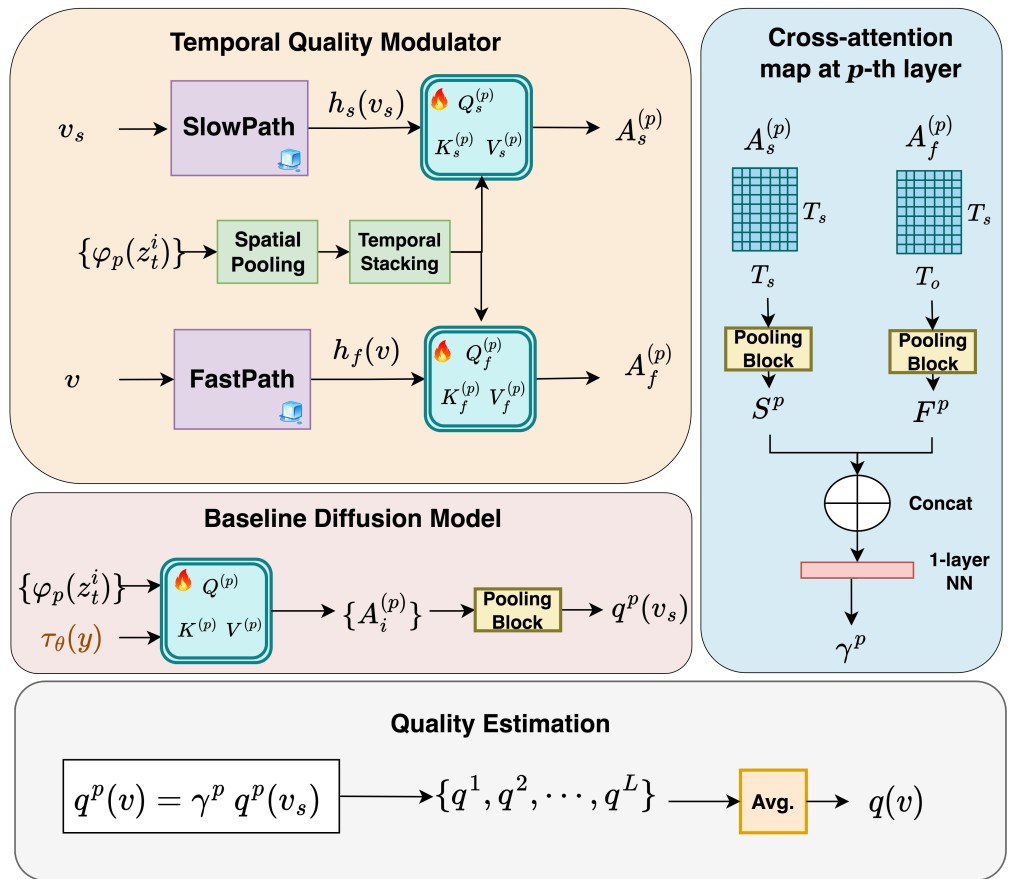

Figure 2: Framework of **Temporal Quality Modulator**. $\{\varphi_p(z_t^i)\}$ is the visual query feature of the UNet at the p$^{th}$ cross-attention block for a time-step $t$ across all sub-sampled frames $i \in \{1, 2, \cdots, T_s\}$. Slow-pathway and fast-pathway features $h_s(v_s)$ and $h_f(v)$ are extracted from frozen slow and fast pathway of pre-trained SlowFastNet. Temporal correction factor $\gamma_p$ is obtained by average pooling visual-motion cross-attention maps $A_s^{(p)}$ and $A_f^{(p)}$ across spatial dimension, then concatenating them and passing them through a single-layer neural network.

## 4 Experiments

### 4.1 Training Details:

We choose Stable Diffusion v2 (Rombach et al., 2022) model (SDM) pretrained on LAION-5B (Schuhmann et al., 2022) dataset with 1.45 billion parameters as our default LDM. We choose the model where the VQ-VAE (Razavi et al., 2019) takes in $512 \times 512$ image resolution. Thus, we resize the images and videos to $512 \times 512$ and process the images through the LDM. We freeze all parameters except the weight matrices of all the **16** cross-attention blocks in SDM-v2 as our goal is to adapt the cross-attention for the QA task. We train GenzIQA with MSE loss for 10 epochs with a batch size of 16 and Adam (Kingma & Ba, 2014) optimizer. We choose the timestep $t$ in the range $(0-100]$, and $\lambda = 0.14$ as default based on 7K images of the official validation set of FLIVE (Ying et al., 2020b). Note that we refer to the noise variance in Eq. 1 through timestep $t$, implicitly referring to $\beta_t$. We train GenzVQA for 6 epochs using similar training parameters as GenzIQA. Since SDM is a T2I model, keeping the query weight frozen while learning GenzIQA is beneficial while for GenzVQA we train query weights along with key and value weights. (Analysis is given in Appendix A.2.)

Table 1: Cross-database performance analysis of **GenzIQA** with other NR-IQA methods. All the methods are trained on **official FLIVE train** set and tested across various IQA databases. ∗ method does not have publicly available training code to benchmark on all databases. Bold and underlined numbers denote the best and second-best performance, respectively.

| Methods | KonIQ-10K | | CLIVE | | PIPAL | | NNID | | CSIQ | | LIVE-IQA | | Average | |
|---|---|---|---|---|---|---|---|---|---|---|---|---|---|---|
| | SRCC | PLCC | SRCC | PLCC | SRCC | PLCC | SRCC | PLCC | SRCC | PLCC | SRCC | PLCC | SRCC | PLCC |
| TReS | 0.669 | 0.710 | 0.729 | 0.714 | 0.362 | 0.359 | 0.805 | 0.794 | 0.587 | 0.517 | 0.543 | 0.445 | 0.612 | 0.589 |
| HyperIQA | 0.589 | 0.635 | 0.636 | 0.660 | 0.304 | 0.327 | 0.658 | 0.651 | 0.497 | 0.428 | 0.514 | 0.438 | 0.519 | 0.523 |
| MetaIQA | 0.578 | 0.489 | 0.448 | 0.410 | 0.340 | 0.312 | 0.452 | 0.429 | 0.562 | 0.536 | 0.732 | 0.673 | 0.486 | 0.479 |
| MUSIQ | 0.648 | 0.692 | 0.662 | 0.687 | 0.341 | 0.331 | 0.776 | 0.778 | 0.484 | 0.583 | 0.259 | 0.335 | 0.582 | 0.567 |
| CLIP-IQA$^+$ | 0.724 | 0.756 | 0.657 | 0.673 | 0.271 | 0.293 | 0.694 | 0.702 | 0.591 | 0.617 | 0.611 | 0.617 | 0.592 | 0.609 |
| Re-IQA | 0.764 | 0.787 | 0.699 | 0.711 | 0.245 | 0.266 | 0.838 | 0.828 | 0.324 | 0.381 | 0.304 | 0.338 | 0.542 | 0.552 |
| ARNIQA | 0.766 | 0.768 | 0.707 | 0.729 | 0.362 | 0.373 | 0.782 | 0.762 | 0.482 | 0.508 | 0.498 | 0.485 | 0.601 | 0.604 |
| GRepQ | **0.781** | 0.786 | 0.736 | 0.753 | 0.303 | 0.318 | 0.843 | 0.832 | 0.579 | 0.587 | 0.666 | 0.568 | 0.638 | 0.641 |
| QCN | 0.732 | 0.783 | 0.724 | 0.767 | 0.370 | 0.382 | 0.814 | 0.808 | 0.599 | 0.671 | **0.806** | **0.779** | 0.652 | 0.698 |
| DP-IQA* | 0.771 | - | 0.770 | - | - | - | - | - | - | - | - | - | - | - |
| PFD-IQA* | 0.775 | - | 0.783 | - | - | - | - | - | - | - | - | - | - | - |
| **GenzIQA** | 0.779 | **0.823** | **0.799** | **0.829** | **0.473** | **0.496** | **0.897** | **0.878** | **0.636** | **0.677** | 0.789 | 0.712 | **0.710** | **0.736** |

During training, we randomly choose a **single timestep value** in the $(0-100]$ range for quality estimation. During inference, we chose a fixed timestep of 50 based on the GenzIQA's validation performance on official FLIVE Ying et al. (2020a) validation split. We experimented with other time-step values in the range $[10-90]$ and found that there is little variation in the validation performance. We evaluate both the models using the Spearman's Rank Order Correlation Co-efficient (SRCC) and the Pearson's Linear Correlation Co-efficient (PLCC) between the predicted quality and ground-truth human opinion scores. All experiments were conducted on a 24 GB NVIDIA RTX 3090 GPU with Pytorch 1.13.

## 4.2 Cross Database IQA Generalization

To study the generalizability of GenzIQA, we train it with the largest user generated content (UGC) dataset, specifically the official FLIVE (Ying et al., 2020a) train database comprising of 30,253 images, and test on diverse categories of distortions. In particular, we evaluate GenzIQA on various categories of test datasets such as **camera-captured** images (KonIQ-10K (Hosu et al., 2020), CLIVE (Ghadiyaram & Bovik, 2015)), **GAN-restored** images (PIPAL (Gu et al., 2020)), **night-time** images (NNID (Hu et al., 2021)), and **synthetically distorted** images (CSIQ (Larson & Chandler, 2010) and LIVE-IQA (Sheikh et al., 2006)). We compare with popular state-of-the-art NR-IQA methods in literature such as TReS (Golestaneh et al., 2022), HyperIQA (Su et al., 2020), MetaIQA (Zhu et al., 2020), MUSIQ (Ke et al., 2021), CLIP-IQA$^+$ (Wang et al., 2023a), ARNIQA (Agnolucci et al., 2024), GRepQ (Srinath et al., 2024), Re-IQA (Saha et al., 2023), QCN (Shin et al., 2024), DP-IQA (Fu et al., 2024), and PFD-IQA Li et al. (2025). We note that all these methods are also trained on the official FLIVE train set for a fair comparison. CLIP-IQA$^+$ is an interesting comparison to GenzIQA as it is a vision-language model where learnable prompts are used to estimate quality from the visual and text features. Since LIQE (Zhang et al., 2023a), Q-Align (Wu et al., 2023d) requires detailed annotations of image-text context, we are not able to benchmark them due to the absence of such annotations on the FLIVE (Ying et al., 2020b) dataset. In Tab. 1, we compare with popular state-of-the-art NR-IQA methods in literature and observe that GenzIQA consistently outperforms other methods across various databases. Further, the performance on most databases is acceptable for IQA.

## 4.3 Cross Database VQA Generalization

We train GenzVQA with the largest UGC dataset, specifically, the official LSVQ (Ying et al., 2021) train database comprising 28,056 videos to evaluate its generalizabilty. In Tab. 2, we evaluate our model against domain-specific videos such as **frame-rate variation** (LIVE-YT-HFR (Madhusudana et al., 2021)), **Ultra-HD** (Waterloo-IVC-4K (Li et al., 2019b)), **gaming** (LIVE-YT-Gaming (Yu et al., 2022) and CGVDS (Zadtootaghaj et al., 2020)), and **streaming** (CSIQ-VQD (Vu & Chandler, 2014) and MD-VQA (Zhang et al., 2023c)) videos. In Tab. 3, we validate against popular VQA databases having **camera-captured** and **UGC** videos such as KoNViD-1K (Hosu et al., 2017), LIVE-VQC (Sinno & Bovik, 2019), Youtube-UGC (Wang

Table 2: Cross-database performance analysis of **GenzVQA** with other NR-VQA methods on high frame-rate, Ultra-HD, gaming, and streaming videos. All methods are trained on **official LSVQ train** set. Bold and underline denote best and second best.

| Methods | LIVE-YT-HFR | | Waterloo-IVC-4K | | LIVE-YT-Gaming | | CGVDS | | CSIQ-VQD | | MD-VQA | |
|---|---|---|---|---|---|---|---|---|---|---|---|---|
| | SRCC | PLCC | SRCC | PLCC | SRCC | PLCC | SRCC | PLCC | SRCC | PLCC | SRCC | PLCC |
| VSFA | 0.461 | 0.528 | 0.465 | 0.487 | 0.658 | 0.721 | 0.718 | 0.734 | 0.497 | 0.502 | 0.589 | 0.651 |
| CSVT-BVQA | 0.351 | 0.422 | 0.365 | 0.407 | 0.631 | 0.673 | 0.791 | 0.811 | 0.580 | 0.581 | 0.626 | 0.652 |
| SimpleVQA | 0.378 | 0.409 | 0.382 | 0.414 | 0.666 | 0.724 | 0.804 | 0.801 | 0.599 | 0.572 | 0.654 | 0.678 |
| CONVIQT | 0.321 | 0.403 | 0.358 | 0.385 | 0.572 | 0.618 | 0.791 | 0.811 | 0.580 | 0.581 | 0.633 | 0.665 |
| DOVER | 0.355 | 0.465 | 0.369 | 0.419 | 0.651 | 0.730 | 0.694 | 0.744 | 0.594 | 0.598 | 0.708 | 0.690 |
| ModularVQA | 0.350 | 0.427 | 0.404 | 0.456 | **0.685** | **0.740** | 0.722 | 0.775 | 0.520 | 0.524 | 0.588 | 0.617 |
| **GenzVQA** | **0.644** | **0.662** | **0.493** | **0.562** | 0.616 | 0.692 | **0.822** | **0.839** | **0.694** | **0.707** | **0.724** | **0.721** |

Table 3: Cross-database performance analysis of **GenzVQA** with other NR-VQA methods on camera-captured and UGC videos. All methods are trained on **official LSVQ train** set. Average corresponds to the mean performance across all datasets in Tab. 2 and 3. Bold and underlined denote best and second best, respectively.

| Methods | LIVE-VQC | | KoNViD-1K | | Youtube-UGC | | LIVE-Qualcomm | | Maxwell | | Average | |
|---|---|---|---|---|---|---|---|---|---|---|---|---|
| | SRCC | PLCC | SRCC | PLCC | SRCC | PLCC | SRCC | PLCC | SRCC | PLCC | SRCC | PLCC |
| VSFA | 0.753 | 0.795 | 0.810 | 0.811 | 0.718 | 0.721 | 0.438 | 0.434 | 0.649 | 0.654 | 0.614 | 0.639 |
| CSVT-BVQA | 0.793 | 0.811 | 0.843 | 0.835 | 0.802 | 0.792 | 0.520 | 0.568 | 0.594 | 0.588 | 0.627 | 0.649 |
| SimpleVQA | 0.749 | 0.789 | 0.826 | 0.820 | 0.802 | 0.806 | 0.570 | 0.617 | 0.697 | 0.704 | 0.647 | 0.667 |
| CONVIQT | 0.706 | 0.737 | 0.775 | 0.782 | 0.715 | 0.704 | 0.534 | 0.613 | 0.645 | 0.652 | 0.603 | 0.632 |
| DOVER | **0.832** | **0.855** | 0.884 | 0.883 | 0.777 | 0.792 | 0.668 | 0.704 | 0.728 | 0.730 | 0.660 | 0.691 |
| ModularVQA | 0.806 | 0.844 | 0.878 | 0.884 | 0.788 | 0.804 | 0.573 | 0.597 | 0.730 | 0.737 | 0.640 | 0.673 |
| **GenzVQA** | 0.826 | 0.840 | **0.885** | **0.888** | **0.824** | **0.829** | **0.707** | **0.719** | **0.745** | **0.757** | **0.725** | **0.747** |

et al., 2019), Maxwell (Wu et al., 2023c), and LIVE-Qualcomm (Ghadiyaram et al., 2018). We compare GenzVQA with state-of-the-art VQA methods and see that our model achieves a consistently superior performance across various video content and distortions. In particular, our gains are substantial for high-frame rate VQA, showing the generalizability of GenzVQA.

We compare GenzVQA with only learning-based methods due to their superior performance with respect to the classical methods as noted in literature (Wu et al., 2023b; Lu et al., 2024). Due to the computational complexity of learning end-to-end VQA methods, earlier works such as VSFA (Li et al., 2019a), CSVT-BVQA (Li et al., 2022), SimpleVQA (Sun et al., 2022), and CONVIQT (Madhusudana et al., 2022a) only learn a regressor with pre-trained video representations. On the other hand, DOVER (Wu et al., 2023b) is an end-to-end VQA method. Since DOVER is an improved version of FAST-VQA (Wu et al., 2022), we do not compare with FAST-VQA. We also include a recent vision-language based VQA model viz. ModularVQA. We note that GenzVQA achieves a consistently superior performance against existing VQA methods across various video content and distortions. This experiment establishes the generalizability of GenzVQA against existing methods. Note, that the mean opinion scores used for training and evaluation for every IQA and VQA datasets are publicly available with the respectively datasets.

## 4.4 Ablation Studies and Detailed Experiments

### 4.4.1 Role of Noise in Latent Diffusion Model for Quality Estimation

The noise added to the latent features $z_0$ can have a significant impact on the ability of the latent diffusion model (LDM) to predict image or video frames' quality. Recall that the denoising UNet of LDM estimates the Gaussian noise incorporated in the forward process. We hypothesize that there is a delicate relationship between the denoising ability of the UNet and the amount of additive noise, which could alter the semantic information and quality information in the latent representation. We investigate this by gradually distorting the original image latent $z_0$ for a fixed length Markov chain. In particular, we generate the images using

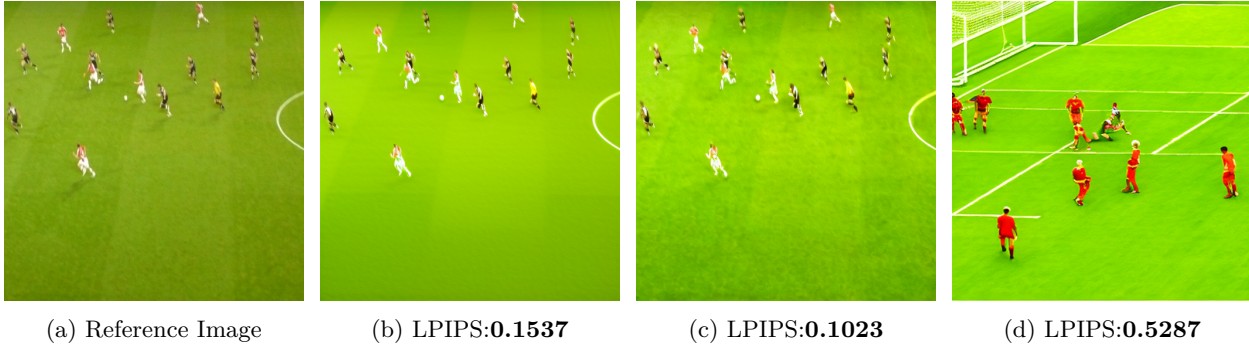

(a) Reference Image      (b) LPIPS:**0.1537**      (c) LPIPS:**0.1023**      (d) LPIPS:**0.5287**

Figure 3: Generated images from zero shot SDM. In Fig. 3b image is generated without noise infused to the image latent, Fig. 3c and Fig. 3d are generated images with noise fed at the timestep $t = 95$ (low noise), and $t = 950$ (high noise) respectively and subsequently denoised. Lower LPIPS scores correspond to better perceptual quality.

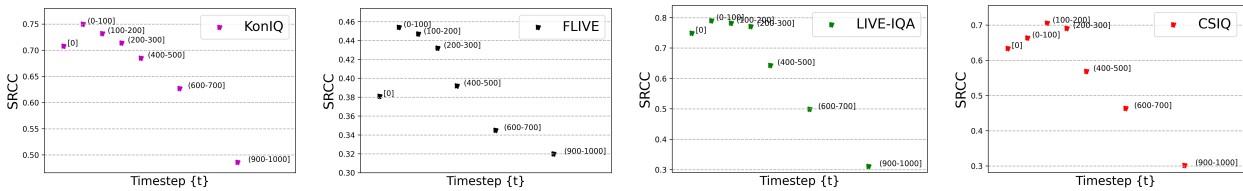

Figure 4: SRCC performance variation of GenzIQA trained on **CLIVE** and tested across four databases at different timesteps.

the pretrained Stable Diffusion v2 (Rombach et al., 2022) model as our default LDM (as mentioned in implementation details of main paper) with various noise steps in forward diffusion process.

In Fig. 3, we generate images from clean, low noise ($t = 95$) and high noise ($t = 950$) versions of the original image features. The generated image from the clean latent $z_0$ in Fig. 3b, is blurry as can be seen from the textureless outfield. This effect maybe attributed to the fact that denoising UNet removes bandpass texture information.

In Fig. 3d, we see that the addition of high Gaussian noise distorts the semantic information in the latent space and thus the UNet generates a content different from the original image. Finally, in Fig. 3c, the image generated preserves both the semantic and texture information as evident visually and from the respective LPIPS (Zhang et al., 2018a) scores. In our study, addition of the correct range of noise is extremely important as we wish to capture the perceptual and semantic information at all intermediate stages of the UNet.

### 4.4.2 Impact of Noise Level Variation

The level of noise added to the image latent space has a direct impact on the ability of the denoiser to preserve perceptual information. Thus, we train GenzIQA on the CLIVE database with varying levels of noise added to the input. Specifically, we train on CLIVE and test on various dataset for a single-timestep sampled in the range $\{0, (0-100], (100-200], (200-300], (400-500], (600-700], (900-1000]\}$. As evident from Fig. 4, the performance across various test datasets is fairly consistent in the range $(0-300]$, while it starts to drastically degrade for noisy timesteps beyond 400. We conclude that corrupting the image latent with high noise distorts the semantic information, thus hindering the extraction of quality relevant information from the cross-attention map. Further, extremely low noise levels cause blur during denoising, leading to poorer quality prediction performance. We see that sampling a single-timestep from the $(0, 100]$ range offers a reasonable performance across all datasets.

Table 4: Impact of various components of **GenzIQA** trained on **CLIVE** and tested on four datasets. We report the SRCC performance.

| Prompt Tuning | Vision -Text Cross- Attn. | LSE Pool | Test Data | | | |
|---|---|---|---|---|---|---|
| | | | KonIQ test | FLIVE test | NNID | LIVE- IQA |
| × | × | × | 0.184 | 0.010 | 0.032 | 0.124 |
| ✓ | × | ✓ | 0.455 | 0.284 | 0.517 | 0.636 |
| × | ✓ | ✓ | 0.696 | 0.331 | 0.677 | 0.706 |
| ✓ | ✓ | × | 0.735 | 0.438 | 0.721 | 0.773 |
| ✓ | ✓ | ✓ | **0.750** | **0.454** | **0.738** | **0.782** |

Table 5: Impact of various components of GenzVQA evaluated on four datasets. We report the SRCC performance.

| Train Data | SlowFast Feature | Vision- Motion Cross- Attn. | Test Data | | | |
|---|---|---|---|---|---|---|
| | | | YT- UGC | LIVE- VQC | YT- HFR | CSIQ- VQD |
| FLIVE | × | × | 0.495 | 0.673 | 0.358 | 0.394 |
| LSVQ | × | × | 0.806 | 0.768 | 0.494 | 0.583 |
| LSVQ | ✓ | × | 0.809 | 0.775 | 0.538 | 0.666 |
| LSVQ | ✓ | ✓ | **0.824** | **0.826** | **0.644** | **0.694** |

### 4.4.3    Impact of major components of GenzIQA

We evaluate the need of cross-attention finetuning between images and textual prompts, learning quality-aware prompts, and LSE pooling (versus average pooling). In Tab. 4, we train GenzIQA on CLIVE and test on $KonIQ_{test}$, $FLIVE_{test}$, NNID, and LIVE-IQA. The zero-shot performance reported in the first row indicates that all components of GenzIQA are necessary to adapt LDM for IQA. Learning the cross-attention map has the maximum impact on performance. However, prompt tuning and LSE pooling also lead to consistent improvements across databases.

### 4.4.4    Impact of major components of GenzVQA

We perform experiments in Tab. 5 with respect to GenzVQA. We first observe that training the baseline diffusion model on a VQA dataset such as LSVQ gives a significant improvement over training it on an IQA dataset such as FLIVE.

**a) Impact of SlowFast features:** One of GenzVQA's main components is the temporal quality modulator. Recent works such as CSVT-BVQA, SimpleVQA, and ModularVQA use pre-trained SlowFast features and regress directly against quality. Thus, we replace our TQM with a two-layer neural network that regresses SlowFast features directly to get the temporal quality correction factor. We notice a gain in performance while using the SlowFast features along with the baseline diffusion model.

**b) Impact of Vision-Motion Cross-Attention module:** A significant gain in performance is observed with the inclusion of the cross-attention module in TQM with respect to the baseline model across all datasets. In particular, cross-attention between the SlowFast motion features and the UNet visual features is advantageous over merely using SlowFast features. This experiment establishes the importance of learning shared information between the UNet and SlowFast networks for TQM.

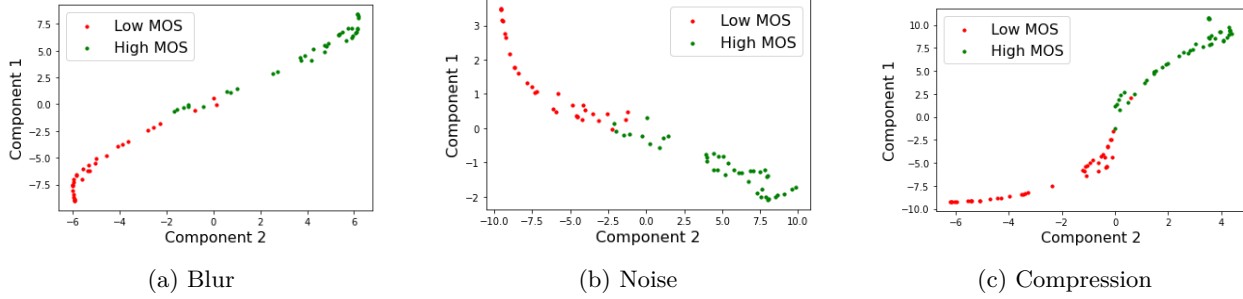

|  | (a) Blur | (b) Noise | (c) Compression |

Figure 5: 2D tSNE visualization of cross-attention features of GenzIQA trained on FLIVE and tested on (a) Gaussian blur, (b) White noise, and (c) JPEG compressed images from LIVE-IQA.

Table 6: Performance comparison of GenzVQA with and without Fast and Slow cross-attention blocks. Video 1 and 2 are selected from LIVE-VQC (Sinno & Bovik, 2019) dataset.

|  | Video 1 | Video 2 |
|---|---|---|
|  |  |  |
| MOS | 85.207 | 18.616 |
| Baseline | 56.894 | 52.153 |
| Mean Fast Score | 1.22 | 0.9136 |
| Mean Slow Score | 0.9312 | 1.3798 |
| GenzVQA | 75.225 | 36.9243 |

### 4.4.5 Impact of VAE on Distortions

Our GenzIQA's backbone viz. SDM applies a diffusion model on the latent space of VQ-VAE. In this experiment, we analyze the impact of both image resizing and VQ-VAE on the image/video frame distortions. In particular, we show the 2D tSNE (Tu et al., 2021b) visualization of the cross-attention representations averaged across all blocks of the UNet model. In Figure 5, we show the features for (a) JPEG compression, (b) white noise, and (c) Gaussian blur images from LIVE-IQA belonging to high MOS ($> 70$) and low MOS($< 40$). We infer that for all distortion types, the diffusion features can segregate images based on their distortion levels. Thus, we observe the distortion information is reasonably preserved despite the resizing and use of VQ-VAE for effective quality assessment.

### 4.4.6 Impact of Slow and Fast Attention Module

In Table 6, we compare the prediction accuracy given by GenzVQA for two videos. For Video 1, the baseline prediction without any fast and slow cross-attention is much lesser than the actual ground-truth. The baseline prediction gets amplified by the ratio of fast to slow cross-attention score to GenzVQA's prediction. Note that we compute fast and slow cross-attention at every scale of Stable diffusion's UNet and get the GenzVQA prediction as an average over all scales as given in Equation 9 in main paper. But for ease of understanding here, we provide the average of fast and slow cross-attention scores across all scales. Similarly, for Video 2, the baseline prediction is much higher than ground-truth. Thus, the baseline gets dampened by the ratio of fast and slow cross-attentions scores.

Table 7: Cross-database performance analysis of **GenzIQA** with Stable diffusion v1.5 and v2 as backbone. All the methods are trained on **official FLIVE train** set and tested across various IQA databases.

| Methods | KonIQ-10K | | CLIVE | | PIPAL | | NNID | | CSIQ | | LIVE-IQA | |
|---|---|---|---|---|---|---|---|---|---|---|---|---|
| | SRCC | PLCC | SRCC | PLCC | SRCC | PLCC | SRCC | PLCC | SRCC | PLCC | SRCC | PLCC |
| ARNIQA | 0.766 | 0.768 | 0.707 | 0.729 | 0.362 | 0.373 | 0.782 | 0.762 | 0.482 | 0.508 | 0.498 | 0.485 |
| QCN | 0.732 | 0.783 | 0.724 | 0.767 | 0.370 | 0.382 | 0.814 | 0.808 | 0.599 | 0.671 | 0.806 | 0.779 |
| GenzIQA (SD v1.5) | 0.772 | 0.803 | 0.785 | 0.793 | 0.452 | 0.465 | 0.874 | 0.848 | 0.594 | 0.651 | 0.772 | 0.717 |
| GenzIQA (SD v2) | 0.779 | 0.823 | 0.799 | 0.829 | 0.473 | 0.496 | 0.897 | 0.878 | 0.636 | 0.677 | 0.789 | 0.712 |

Table 8: Performance comparison of **GenzIQA** with other NR-IQA methods on Intra-database setting. All results are obtained from from their respective publications.

| Methods | FLIVE | | KonIQ-10K | | CLIVE | | LIVE-IQA | | SPAQ | | CSIQ | |
|---|---|---|---|---|---|---|---|---|---|---|---|---|
| | SRCC | PLCC | SRCC | PLCC | SRCC | PLCC | SRCC | PLCC | SRCC | PLCC | SRCC | PLCC |
| TReS | 0.554 | 0.625 | 0.915 | 0.928 | 0.846 | 0.877 | 0.969 | 0.968 | - | - | 0.922 | 0.942 |
| HyperIQA | 0.535 | 0.623 | 0.906 | 0.917 | 0.859 | 0.882 | 0.962 | 0.966 | 0.916 | 0.919 | 0.923 | 0.942 |
| DB-CNN | 0.554 | 0.652 | 0.875 | 0.884 | 0.851 | 0.869 | 0.968 | 0.971 | 0.911 | 0.915 | 0.946 | 0.959 |
| CONTRIQUE | 0.580 | 0.651 | 0.894 | 0.904 | 0.845 | 0.857 | 0.960 | 0.961 | 0.914 | 0.919 | 0.942 | 0.955 |
| Re-IQA | 0.645 | 0.733 | 0.914 | 0.923 | 0.840 | 0.854 | 0.970 | 0.971 | 0.918 | 0.925 | 0.947 | 0.960 |
| LIQE | - | - | 0.918 | 0.908 | 0.889 | 0.879 | 0.958 | 0.942 | - | - | 0.923 | 0.918 |
| ARNIQA | 0.595 | 0.671 | - | - | - | - | 0.966 | 0.970 | 0.905 | 0.910 | 0.962 | 0.973 |
| GRepQ | 0.531 | 0.582 | 0.908 | 0.916 | 0.859 | 0.867 | 0.945 | 0.943 | 0.874 | 0.877 | 0.948 | 0.955 |
| QPT | 0.645 | 0.733 | 0.927 | 0.941 | **0.895** | **0.914** | - | - | 0.925 | 0.928 | - | - |
| QCN | 0.644 | **0.741** | 0.934 | 0.945 | 0.875 | 0.893 | - | - | 0.923 | 0.928 | - | - |
| LoDA | 0.578 | 0.679 | 0.932 | 0.944 | 0.876 | 0.899 | **0.975** | **0.979** | 0.925 | 0.928 | - | - |
| DSMix | **0.646** | **0.735** | 0.915 | 0.925 | 0.873 | 0.883 | 0.974 | 0.974 | - | - | 0.957 | 0.962 |
| **GenzIQA** | 0.627 | 0.728 | **0.936** | **0.950** | 0.879 | 0.897 | 0.966 | 0.968 | **0.929** | **0.935** | **0.968** | **0.972** |

### 4.4.7 Impact of baseline Latent Diffusion Model

In this experiment, we explore the cross-database generalizability of GenzIQA for a different variant of Stable diffusion (SD) models. In Tab. 7, we train GenzIQA with SD-v1.5 as backbone on official FLIVE train dataset similar to Sec. 4.2 and test on remaining IQA datasets. We infer that GenzIQA achieves good cross-dataset generalization for both the variants of SDMs. The performance of SD-v2 variant is better than SD-v1.5 variant due to the superior text-encoder viz. OpenCLIP ViT-H/14 over OpenCLIP Vit-L/14. Also, the text encoder is trained with LAION dataset in v2, while in v1.5 it is a frozen CLIP model.

### 4.4.8 Intra-database Performance

**a) Intra-database Performance of GenzIQA:** We now validate the performance of our model in intra-database train-test settings. Specifically, we train GenzIQA with either the official train set or 80% of image samples from various databases and test on the official test set or remaining 20%, respectively. In Tab. 8, in case of FLIVE and KonIQ-10K, we train-test on the official split provided, while for other datasets, we randomly split the data 10 times in the ratio 80 : 20 and report the median performance. We compare GenzIQA with other NR-IQA methods in the same setting. We also benchmark LIQE (Zhang et al., 2023a) by training it on individual datasets (wherever detailed text annotation was provided). We infer that our method gives competitive performance with recent state-of-the-art methods across all databases. We conclude from Tab. 8 that GenzIQA not only outperforms recent benchmarks in practical cross/inter database generalization scenarios but also does remarkably well on intra-database test scenarios.

**b) Transfer Learning GenzVQA on Smaller Datasets:** Similar to other works such as FAST-VQA (Wu et al., 2022), DOVER (Wu et al., 2023b), and ModularVQA (Wen et al., 2024), we finetune GenzVQA on individual evaluation datasets viz. LIVE-VQC (Sinno & Bovik, 2019), KoNViD-1K (Hosu et al., 2017), and Youtube-UGC (Wang et al., 2019), LIVE-Qualcomm (Ghadiyaram et al., 2018) and LIVE-YT-Gaming (Yu et al., 2022) in 80 : 20 train-test ratio for 10 splits and report the median performance. In case of LSVQ (Ying et al., 2021), we train on the official train split and report the performances on the official test

Table 9: Finetune performance comparison of **GenzVQA** with other NR-VQA methods on various databases. KSVQE results are from (Lu et al., 2024) and all other methods numbers are taken from ModularVQA.

| Methods | LSVQ-test | | LSVQ-1080p | | LIVE-VQC | | KoNViD-1K | | Youtube-UGC | | LIVE-Qualcomm | | LIVE-YT-Gaming | |
|---|---|---|---|---|---|---|---|---|---|---|---|---|---|---|
| | SRCC | PLCC | SRCC | PLCC | SRCC | PLCC | SRCC | PLCC | SRCC | PLCC | SRCC | PLCC | SRCC | PLCC |
| VSFA | 0.801 | 0.796 | 0.675 | 0.704 | 0.718 | 0.771 | 0.794 | 0.799 | 0.787 | 0.789 | 0.708 | 0.774 | 0.784 | 0.819 |
| CSVT-BVQA | 0.852 | 0.854 | 0.772 | 0.788 | 0.841 | 0.839 | 0.839 | 0.830 | 0.825 | 0.818 | 0.833 | 0.837 | 0.852 | 0.868 |
| SimpleVQA | 0.866 | 0.863 | 0.750 | 0.793 | 0.740 | 0.775 | 0.792 | 0.798 | 0.819 | 0.817 | 0.722 | 0.774 | 0.814 | 0.836 |
| FastVQA | 0.876 | 0.877 | 0.779 | 0.814 | 0.853 | 0.873 | 0.893 | 0.887 | 0.863 | 0.859 | 0.807 | 0.814 | 0.869 | 0.880 |
| DOVER | 0.888 | 0.889 | 0.795 | 0.830 | 0.853 | 0.872 | 0.892 | 0.900 | 0.875 | 0.874 | 0.736 | 0.789 | 0.882 | 0.906 |
| ModularVQA | 0.895 | 0.895 | 0.809 | 0.844 | 0.860 | 0.880 | 0.901 | 0.905 | 0.876 | 0.877 | 0.832 | 0.842 | 0.867 | 0.902 |
| KSVQE | 0.886 | 0.888 | 0.790 | 0.823 | 0.861 | 0.883 | **0.922** | **0.921** | 0.900 | 0.912 | - | - | - | - |
| **GenzVQA** | **0.898** | **0.899** | **0.797** | **0.835** | **0.871** | **0.882** | 0.909 | 0.918 | **0.910** | **0.913** | **0.851** | **0.856** | **0.878** | **0.907** |

and 1080$p$ splits. We compare GenzVQA with other popular NR-VQA in Tab. 9. We infer that GenzVQA outperforms the state-of-the art methods on most databases.

### 4.4.9 Analyzing Cross-Attention Map Representation

We visualize the cross-attention representation of GenzIQA trained on FLIVE to understand why it leads to superior generalization. For this, we subsample good and bad quality images from three different test datasets with MOS less than 30 and greater than 70, respectively. In Fig. 6, we show the 2D tSNE (Tu et al., 2021b) visualization of the cross-attention map in Eq. (2) averaged across all blocks, timestep samples and the number of image tokens. In particular, we chose a perplexity of 40 and iterated the optimization over 1000 steps while generating the tSNE plot. We conclude that the learned attention representation clearly separates high and low-quality images as evident from Fig. 6. Under similar settings, we also visualize the representation of CLIP-IQA$^+$ visual encoder conditioned on the same prompt pair attributes. In particular, the visual feature similarity with ensemble text representation gives a 2-dimensional representation for each image. We see that the CLIP-IQA$^+$ model's separability is somewhat inferior to what we see with the cross-attention features of GenzIQA.

### 4.4.10 Analysis of Contextual Prompt Learning

**a) Study on Learnable Prompt vs Fixed Prompt:**

GenzIQA and GenzVQA by default are trained with learnable antonym prompts using CoOp (Zhou et al., 2022) similar to CLIP-IQA$^+$. We study the need for diverse prompts pairs as well as the need for learnable context vectors vs fixed prompts in case of GenzIQA. In Tab. 10, we choose *['Good Photo.', 'Bad Photo.']* as the initial antonym prompt pair while for the single prompt case, we only choose *['Good Photo.']*. We make two observations from this study. Firstly, antonym prompt pairs give a better performance than a single prompt in both the learnable prompt and fixed prompt training across all test datasets. Secondly, learnable prompts consistently yield superior results with respect to the fixed prompt case. Both these phenomena are expected as multiple studies show the benefit of prompt learning (Yao et al., 2023; Guo et al., 2023; Wang et al., 2023a).

**b) Analysis on Choice of Prompts:** In our experimental studies, we choose *['Good Photo.', 'Bad Photo.']* as our initial learnable antonym prompt attributes. As shown in CLIP-IQA$^+$ (Wang et al., 2023a), this prompt pair gives the best estimate of quality. Here, we train GenzIQA with the official FLIVE training set for initial prompts *['High Quality.', 'Low Quality.']* in Tab. 11, and *['High Definition.', 'Low Definition.']* in Tab. 12 under different settings. We see that there is minimal variation in performance with respect to the exact choice of these popular quality relevant antonym prompt attributes.

### 4.5 Run Time:

The average test-time required by GenzIQA to estimate quality for a single $512 \times 512$ resized image for one timestep on a 24 GB NVIDIA RTX 3090 is **0.035** seconds. While for a 8-second long 30 fps video,

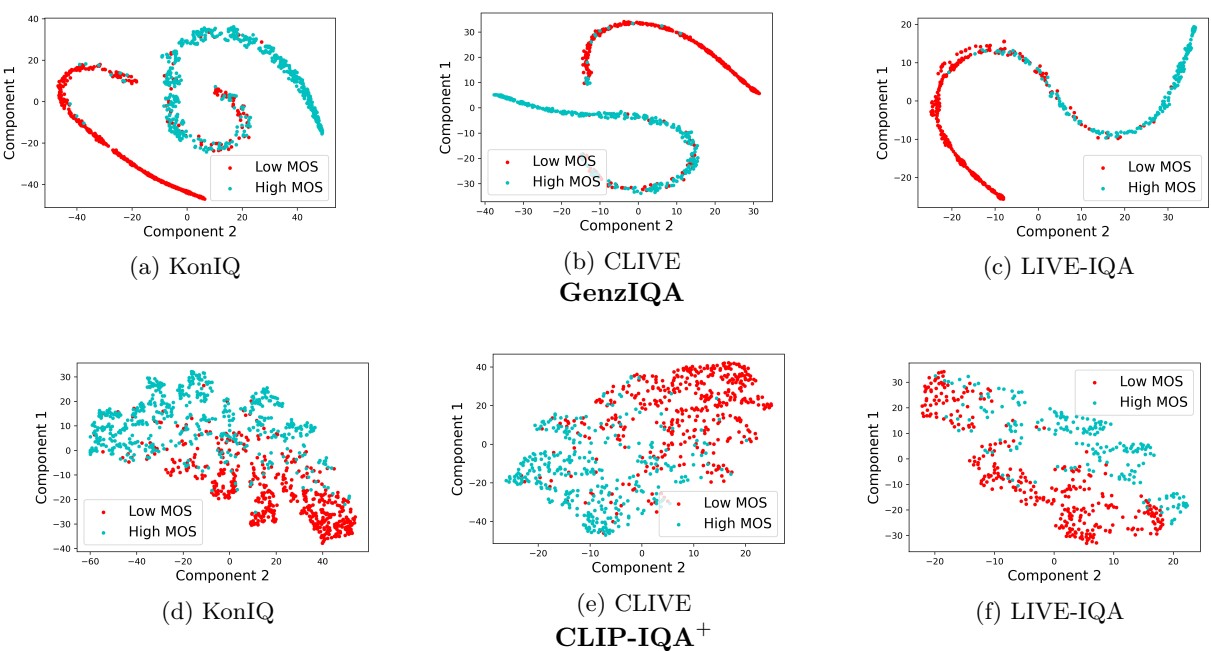

Figure 6: 2D tSNE visualization of cross-attention features of **GenzIQA** trained on **FLIVE** and tested on images from KonIQ, CLIVE, and LIVE-IQA. CLIP-IQA$^+$ similarity features conditioned on antonym prompts are also shown.

Table 10: SRCC performance variation of **GenzIQA** on the choice of prompts being fed as input to the CLIP text encoder. All the variations have been trained on **FLIVE** official train set and cross-tested on these datasets. **Good Photo / Bad Photo** as initial attributes.

| Test database | Trainable single prompt | Trainable antonym prompts | Fixed single prompt | Fixed antonym prompts |
|---|---|---|---|---|
| **CLIVE** | 0.791 | 0.799 | 0.784 | 0.789 |
| **NNID** | 0.880 | 0.897 | 0.871 | 0.874 |
| **CSIQ** | 0.622 | 0.636 | 0.568 | 0.611 |
| **LIVE-IQA** | 0.783 | 0.789 | 0.731 | 0.759 |

GenzVQA takes **0.357** seconds to estimate quality. Similar to FastVQA (Wu et al., 2022) and ModularVQA (Wen et al., 2024), we infer GenzVQA for 8 second long 30 fps 1080p videos on an NVIDIA RTX 3090 GPU. We compare the inference time of GenzVQA with other NR-VQA methods in Tab. 13. We note that

Table 11: SRCC performance variation of **GenzIQA** trained on FLIVE with **High Quality / Low Quality** as initial prompts.

| Test database | Trainable single prompt | Trainable antonym prompts | Fixed single prompt | Fixed antonym prompts |
|---|---|---|---|---|
| **CLIVE** | 0.776 | 0.795 | 0.758 | 0.761 |
| **NNID** | 0.883 | 0.889 | 0.870 | 0.872 |
| **CSIQ** | 0.616 | 0.623 | 0.546 | 0.576 |
| **LIVE-IQA** | 0.782 | 0.812 | 0.734 | 0.742 |

Table 12: SRCC performance variation of **GenzIQA** trained on FLIVE with **High Definition / Low Definition** as initial prompts.

| Test database | Trainable single prompt | Trainable antonym prompts | Fixed single prompt | Fixed antonym prompts |
|---|---|---|---|---|
| **CLIVE** | 0.768 | 0.791 | 0.752 | 0.755 |
| **NNID** | 0.880 | 0.892 | 0.869 | 0.871 |
| **CSIQ** | 0.612 | 0.620 | 0.548 | 0.564 |
| **LIVE-IQA** | 0.780 | 0.800 | 0.728 | 0.738 |

inference time of GenzVQA (0.357 secs) is much lesser than actual duration (8 secs) of the video. We also compare the number of model parameter of GenzVQA with other VQA methods. Even though the number of parameters of GenzVQA is higher than other benchmarking methods, applying the T2I diffusion model at 1 fps keeps the latency during training/ inference within comparable limits.

Table 13: Inference time and Model parameters comparison of GenzVQA with other NR-VQA methods for 8 second long 30 fps 1080p videos.

| Method | Inference Time(sec) | Parameters(millions) |
|---|---|---|
| VSFA | 11.109 | 24.1 |
| CSVT-BVQA | 27.632 | 58.6 |
| SimpleVQA | 0.714 | 58.3 |
| FastVQA | 0.045 | 28.1 |
| DOVER | 0.047 | 56.2 |
| ModularVQA | 0.159 | 87.8 |
| GenzVQA | 0.357 | 950 |

## 5 Conclusion

In this work, we presented GenzIQA and GenzVQA by leveraging the benefits of LDM. Our work is perhaps one of the earliest attempts at understanding whether and how such models can be used for cross-database generalization in NR-IQA and NR-VQA. In this context, it is important to finetune the cross-attention module and learn quality-aware input context vectors to enable the diffusion models be effective for QA. Also, we introduce a cost-effective way to estimate video quality by introducing our TQM. Although our method can be readily deployed on servers or on the cloud, the complexity of a large vision language model can limit its deployment on edge devices. It would be interesting to explore knowledge distillation (Meng et al., 2023) and inference-time acceleration techniques Salimans & Ho (2022) for edge-deployment of GenzIQA and GenzVQA. On the other hand, while GenzIQA achieves very good performance on most datasets, there is still scope for improvement on PIPAL, which requires a more fine-grained approach to IQA. Nevertheless, we believe that GenzIQA and GenzVQA will encourage further studies on the use of generative models for superior and practical QA.

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

# A    Appendix

## A.1    Impact of Denoising Steps on Quality Estimation

In Fig.3 of the main paper, we showed that a moderate noise variance is effective while extracting quality features during a single denoising step of reverse diffusion. We now address the complementary question of whether increasing the number of denoising steps and using the latent features from the output after more denoising steps can yield richer features. To understand this, we conduct an experiment where we let $t \in (0, 100]$ and increase the number of denoising steps from 1 to 5 by reducing $t$ by 20 in each successive step. This reduction of $t$ aligns with how LDM suggests what noise needs to be added in successive denoising steps. We train the **GenzIQA** model on the CLIVE dataset and test on multiple datasets for this experiment. In Tab. 14, we observe that the multi-step denoising performance is always inferior to the single-step denoising performance, and the performance degrades as we tap features from successive denoisers. We believe that since the cross-attention matrices are tuned for quality estimation, this can impact the ability of the denoiser to remove noise for its effective use in successive steps of multi-step denoising. We conclude that it is best to use a single step denoiser to extract quality-aware features from diffusion models.

Table 14: SRCC performance variation of GenzIQA in several steps of a multistep denoising process.

| Test database | 1st denoising step | 3rd denoising step | 5th denoising step |
|---|---|---|---|
| KonIQ$_{test}$ | 0.747 | 0.631 | 0.562 |
| LIVE-IQA | 0.782 | 0.638 | 0.476 |
| NNID | 0.737 | 0.557 | 0.417 |
| CSIQ | 0.661 | 0.496 | 0.415 |

## A.2    Impact of Vision-Language Cross-Attention Components

In implementation details, we kept the weight matrix $W_Q^{(p)}$ of the query for all cross-attention blocks $p \in \{1, 2, \cdots, L\}$ frozen for **GenzIQA** as we want to preserve the robust visual information captured by the pre-trained text-to image (T2I) Stable diffusion model (SDM). The key and value weights are obtained based on the text prompt, the context of which is also learnt. Thus, we update the key and value weights. In this section, we analyze the impact of freezing query, key and value weights on quality estimation. In Tab. 15, we evaluate GenzIQA trained on CLIVE against four different IQA databases viz. the official test set of KonIQ-10K (Hosu et al., 2020), FLIVE (Ying et al., 2020a) and entire NNID (Hu et al., 2021), and LIVE-IQA (Sheikh et al., 2006). We infer from the last row that making query weights trainable has an adverse impact on performance. Similarly, freezing the key and value weights also affects the performance.

In case of **GenzVQA**, since SDM is a T2I model, we find that learning the query weights (in addition to key and value weights like GenzIQA) is beneficial, as seen in Tab. 16. We infer that learning the query weights is important in capturing spatial attributes in the video which are absent in image representations of the pre-trained T2I model.

## A.3    Choice of Sampling Timesteps

In the implementation details of the main paper, we chose a single sampling timesteps during testing for both GenzIQA and GenzVQA. In Fig. 7, we evaluate GenzIQA trained on the official FLIVE (Ying et al., 2020b) training set on KonIQ (Hosu et al., 2020), CLIVE (Ghadiyaram & Bovik, 2015), NNID (Hu et al., 2021), and LIVE-IQA (Sheikh et al., 2006) with respect to the number of sampling steps. We see that as the number of sampling steps during evaluation increases, the performance also marginally increases and saturates at around 4 on all datasets. Thus, we chose a single sampling timestep for faster test-time evaluations.

Table 15: SRCC performance analysis on the impact of various components of Cross-Attention block in GenzIQA trained on **CLIVE** and tested on four datasets.

| Query Weights | Key Weights | Value Weights | KonIQ$_{test}$ | FLIVE$_{test}$ | NNID | LIVE-IQA |
|:---:|:---:|:---:|:---:|:---:|:---:|:---:|
| × | × | × | 0.455 | 0.284 | 0.517 | 0.636 |
| ✓ | × | × | 0.715 | 0.410 | 0.713 | 0.657 |
| × | ✓ | ✓ | **0.750** | **0.454** | **0.738** | **0.782** |
| ✓ | ✓ | ✓ | **0.750** | 0.426 | 0.718 | 0.752 |

Table 16: SRCC performance analysis on the impact of learning query in GenzVQA trained on the official LSVQ training set and evaluated on seven datasets. Key and value are trainable in both the instances.

| Query Weights | LIVE-VQC | KoNVid-1K | YT-UGC | LIVE-QCOMM | Waterloo-4K | YT-Gaming | CSIQ-VQD |
|:---:|:---:|:---:|:---:|:---:|:---:|:---:|:---:|
| × | 0.810 | 0.874 | 0.814 | 0.680 | 0.453 | 0.598 | 0.678 |
| ✓ | **0.826** | **0.885** | **0.824** | **0.707** | **0.493** | **0.616** | **0.694** |

### A.4  Comparison of GenzIQA and LIQE

LIQE (Zhang et al., 2023a) extends CLIP-IQA$^+$ and has shown very promising performance. However it requires detailed text annotations in the form of quality, distortion and scene information for training. Since FLIVE (Ying et al., 2020a) does not have such detailed annotations, we were unable to benchmark it in Tab. 1. Thus, for a fair comparison with LIQE (Zhang et al., 2023a), we train GenzIQA **solely** on the combination of KADID-10K (Lin et al., 2019) and KonIQ-10K (similar to Table 2 of LIQE). We evaluate GenzIQA and LIQE on various IQA datasets such as TID (Ponomarenko et al., 2015), SPAQ (Fang et al., 2020), PIPAL (Gu et al., 2020), CLIVE (Ghadiyaram & Bovik, 2015), LIVE-IQA (Sheikh et al., 2006) (denoted as LIVE) and NNID (Hu et al., 2021) in a cross-dataset generalized setting and present the SRCC performance in Tab. 17. We note that GenzIQA does better than LIQE in most datasets despite not requiring any detailed text annotations.

### A.5  Cross-Database Generalization on AI Generated Images

In this section, we investigate the effectiveness of our approach in cross-database generalization on the AI-Generated Image (AIGI) databases. To explore this, we choose the largest AIGI database currently available, AIGIQA-20K (Li et al., 2024a) and train GenzIQA on it. The official train split is available for the AIGIQA-

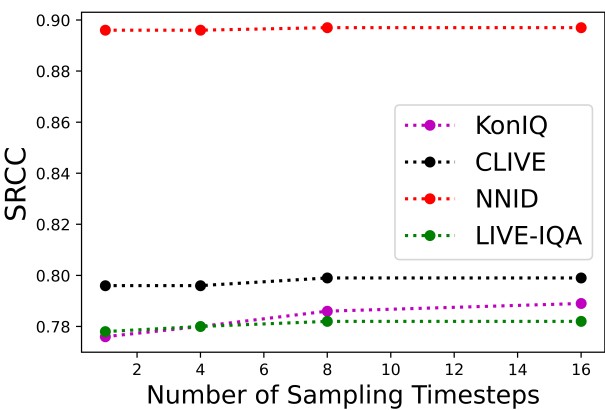

Figure 7: Performance analysis of GenzIQA with varying number of sampling timesteps during evaluation across four test databases.

Table 17: SRCC performance comparison of GenzIQA with LIQE trained on KADID-10K and KonIQ-10K and evaluated across multiple datasets.

| Method | TID | SPAQ | PIPAL | CLIVE | LIVE | NNID |
|---|---|---|---|---|---|---|
| **LIQE** | 0.811 | 0.881 | 0.478 | 0.830 | **0.868** | 0.785 |
| **GenzIQA** | **0.820** | **0.892** | **0.486** | **0.860** | 0.847 | **0.797** |

Table 18: SRCC performance comparison of GenzIQA with other state-of-the-art NR-IQA methods in a cross-database scenario on AI-generated image databases. All models are trained on AIGIQA-20K (Li et al., 2024a), the cross-database evaluation is done for other 3 databases.

| Methods | AGIQA-3K | | AGIQA-1K | | AIGCIQA2023 | |
|---|---|---|---|---|---|---|
| | SRCC | PLCC | SRCC | PLCC | SRCC | PLCC |
| CLIP-IQA$^+$ | 0.498 | 0.486 | 0.268 | 0.266 | 0.475 | 0.459 |
| CONTRIQUE | 0.629 | 0.648 | 0.451 | 0.544 | 0.591 | 0.598 |
| Re-IQA | 0.648 | 0.656 | 0.176 | 0.257 | 0.540 | 0.524 |
| GRepQ | 0.721 | 0.742 | 0.440 | 0.542 | 0.631 | 0.610 |
| IPCE | 0.816 | 0.831 | 0.459 | 0.548 | 0.731 | 0.713 |
| **GenzIQA** | 0.793 | 0.806 | 0.653 | 0.757 | 0.661 | 0.641 |

20K database. For cross-database evaluation, we choose rest of the AIGI databases viz. AGIQA-3K (Li et al., 2023b), AGIQA-1K (Zhang et al., 2023b) and AIGCIQA2023 (Wang et al., 2023b). We benchmark GenzIQA against other state-of-the-art quality representation learning methods such as CONTRIQUE (Madhusudana et al., 2022b), Re-IQA (Saha et al., 2023), GRepQ (Srinath et al., 2024), IPCE (Peng et al., 2024) and report the analysis in Tab. 18. We observe that our method performs consistently well across multiple generative IQA databases.

## A.6 Different Variants of Stable Diffusion

As argued in the implementation details, we choose the input images and video frames resolution to the VQ-VAE as $512 \times 512$. Here, we analyze the impact of our choice in resolution on quality estimation. In Tab. 19, we study the impact on downscaling by comparing the performance at $512 \times 512$ with $256 \times 256$. We train GenzIQA on two different datasets viz. KonIQ-10K (official test-split) and CLIVE and test in a cross-database setting. While training and inference become $4\times$ faster at $256 \times 256$, the performance drastically deteriorates over all the train-test settings. We conclude that the higher resolution model is a better choice for significant performance gains even though the inference time is slower. Again choosing a variant with higher resolution than $512 \times 512$ will require more compute than a 24 GB commercial GPU.

## A.7 Details of Datasets

### A.7.1 Image Quality Assessment Datasets

To validate the generalizable capability of GenzIQA and other NR-IQA methods, we consider various datasets for training and evaluation purpose. These datasets are chosen to cover camera-captured, GAN restored images, night-time captured and synthetically distorted image databases. In Tab. 20 we have given a comprehensive description of these databases.

### A.7.2 Video Quality Assessment Datasets

To train and evaluate GenzVQA and other NR-VQA methods, we perform various experiments on publicly available VQA databases. These databases cover diverse categories of videos, such as user-generated content (UGC), including camera-captured videos, streaming content, high frame rate, ultra-HD, and gaming. Table 21 gives a comprehensive analysis of these databases.

Table 19: SRCC performance comparison of GenzIQA for different Stable Diffusion variants. Stable Diffusion v2 with two VQ-VAE variants feeding $256 \times 256$ and $512 \times 512$ sized images are considered.

| Resolutions | Train on CLIVE | | | | |
|---|---|---|---|---|---|
| | $\mathbf{KonIQ}_{test}$ | **NNID** | **CSIQ** | **LIVE-IQA** | $\mathbf{FLIVE}_{test}$ |
| $256 \times 256$ | 0.653 | 0.725 | 0.567 | 0.629 | 0.362 |
| $512 \times 512$ | 0.750 | 0.738 | 0.664 | 0.782 | 0.454 |
| Resolutions | Train on KonIQ | | | | |
| | **CLIVE** | **NNID** | **CSIQ** | **LIVE-IQA** | $\mathbf{FLIVE}_{test}$ |
| $256 \times 256$ | 0.700 | 0.776 | 0.533 | 0.592 | 0.445 |
| $512 \times 512$ | 0.793 | 0.782 | 0.658 | 0.788 | 0.489 |

Table 20: Summary of publicly available IQA datasets for GenzIQA analysis.

| Dataset | # Images | Resolution | Image Category |
|---|---|---|---|
| FLIVE (Ying et al., 2020a) | 39810 | 160p-700p | Camera-captured |
| KonIQ-10K (Hosu et al., 2020) | 10073 | 768p | Camera-captured |
| CLIVE (Ghadiyaram & Bovik, 2015) | 1162 | 500p-640p | Camera-captured |
| SPAQ (Fang et al., 2020) | 11125 | 1080p-4368p | Camera-captured |
| PIPAL (Gu et al., 2020) | 23200 | 288p | GAN-restored |
| NNID (Hu et al., 2021) | 1340 | 512p | Night-time captured |
| CSIQ (Larson & Chandler, 2010) | 866 | 512p | Synthetically distorted |
| LIVE-IQA (Sheikh et al., 2006) | 779 | 480p-512p | Synthetically distorted |
| TID (Ponomarenko et al., 2015) | 3000 | 384p | Synthetically distorted |

Table 21: Summary of publicly available VQA datasets for GenzVQA analysis.

| Dataset | # Videos | Duration (secs) | Spatial Resolution | Frame Rate | Video Category |
|---|---|---|---|---|---|
| LSVQ (Ying et al., 2021) | 38,811 | 5-12 | $\leq$ 4K | 60 | UGC |
| KoNVid-1K (Hosu et al., 2017) | 1200 | 8 | 540p | 24,25,30 | UGC |
| LIVE-VQC (Sinno & Bovik, 2019) | 585 | 10 | 1080p | 30 | UGC |
| Youtube-UGC (Wang et al., 2019) | 1200 | 20 | 360p-4K | 30 | UGC |
| MaxVQA (Wu et al., 2023c) | 4543 | 4-5 | 240p-1080p | $\leq$ 60 | UGC |
| Live-Qualcomm (Ghadiyaram et al., 2018) | 208 | 15 | 1080p | 30 | UGC |
| Waterloo-IVC -4K (Li et al., 2019b) | 1200 | 9-10 | 540p, 1080p, 4K | 24,25,30 | Ultra-HD |
| LIVE YT-HFR (Madhusudana et al., 2021) | 480 | 6-10 | 1080p | 24,30,60, 82,90,120 | Frame-rate Variation |
| LIVE YT-Gaming (Yu et al., 2022) | 600 | 8-9 | 360p-1080p | 30,60 | Gaming |
| CGVDS (Zadtootaghaj et al., 2020) | 367 | 30 | 480p,720p, 1080p | 20,30,60 | Gaming |
| CSIQ-VQD (Vu & Chandler, 2014) | 180 | 10 | 480p | $\leq$ 60 | Streaming |
| MD-VQA (Zhang et al., 2023c) | 3762 | 8 | 720p, 1080p | $\leq$ 60 | Streaming |

