# OpenReview forum: "Image and Video Quality Assessment using Prompt-Guided Latent Diffusion Models for Cross-Dataset Generalization"
_TMLR — Accepted by TMLR_

### Review · Reviewer_iFTE · 2025-07-24

**Summary Of Contributions:**

The paper proposes a method for assessing the quality of images or videos, based on pretrained text-to-image diffusion models and specifically Stable Diffusion (SD). The core idea for computing quality scores builds on Discffusion (He et al. 2023), which fine-tunes weight matrices in SD’s cross-attention layers between image and text tokens. By applying Log-Sum-Exponential pooling on these cross-attention values, the method derives discriminative quality scores. This cross-attention technique is combined with "Contextual Prompt Tuning", which uses learnable prompts (similar to CoOp (Zhou et al. 2022)) and antonym prompts (inspired by CLIP-IQA). For video quality assessment, the method is applied to temporally subsampled frames to reduce computational cost. To compensate for lost motion information, it integrates SlowFast networks, which compute quality scores from both subsampled and original frames.

The approach is evaluated on cross-database benchmarks for image and video quality assessment, outperforming prior methods. An extensive ablation study further analyzes key aspects of the proposed approach.

**Audience:**

Yes

**Broader Impact Concerns:**

No concerns from my side

**Claims And Evidence:**

Yes

**Requested Changes:**

This work would be strengthened by providing clearer motivations and intuitions behind the key methodological choices (as outlined in the weaknesses above).

**Strengths And Weaknesses:**

Strengths:
- The proposed approach outperforms prior methods, achieving state-of-the-art results on cross-database benchmarks for both image and video quality assessment. It also performs competitively on intra-database benchmarks (matching prior work for images and surpassing it for videos).
- The ablation and experimental analysis are extensive.
- The work shows that cross-attention maps from text-to-image diffusion models—particularly Stable Diffusion—can effectively assess image and video quality. This is the key insight of the paper and one I found interesting.

Weaknesses:
The main weaknesses of the paper is that it does not really explain / discuss the motivation / intuition behind the proposed approach. Instead, it appears to combine prior techniques (Discffusion, CoOp, CLIP-IQA) without justifying their selection.  So, overall the writing and presentation of the paper needs improvement.

- While the image quality assessment approach (Section 3.2) is relatively easy to follow, the motivation behind the design choices is unclear. The paper does not explain why cross-attention maps (via Discffusion) should produce good quality scores or why antonym prompts improve performance. Instead, it appears to combine prior techniques without justifying their selection.

- Similarly, the integration of SlowFast networks with the SD model lacks intuitive discussion (e.g., from the "For TQM..." paragraph onward). The description is very "dry", relying on just providing the equations with minimal explanation of design choices, making it hard to follow. Additionally, Figure 2 is not sufficiently clear or intuitive.

- In the Training Details (subsection 4.1), cross-attention maps are extracted using a random timestep between (0, 100]. The rationale for this randomness—rather than a fixed step (e.g., 50 or 100)—is unexplained. Introducing non-deterministic level of noise during inference seems counterintuitive without justification.

- The use of SlowFast networks is ambiguous: it is unclear whether they are pre-trained, how they were pre-trained, or which layers are used for feature extraction.

---

> ### Author Response · Authors · 2025-08-09
> **Response to Reviewer iFTE.**
>
> We are thankful for all your analysis of our work. We appreciate that you find the use of cross-attention learning for image/ video quality assessment task interesting. Here, we provide a detailed answers to the queries raised.
>
> **Q1) While the image quality assessment approach (Section 3.2) is relatively easy to follow, the motivation behind the design choices is unclear. The paper does not explain why cross-attention maps (via Discffusion) should produce good quality scores or why antonym prompts improve performance. Instead, it appears to combine prior techniques without justifying their selection.**
>
> Ans: **Motivation behind using cross-attention maps for Quality Estimation:**
>
>   (i) Stable Diffusion model (SDM) [1] is trained with image-text pairs from LAION-5B dataset. Such datasets include perceptual attributes/ aspects in its prompts such as detailed descriptions that are related to visual quality, and aesthetic-style captioning. Moreover, the 5 billion dataset size encompasses diverse content and quality.  While generating high-quality images from noisy latent features using text-guided prompts, the generative pipeline progressively retrieves perceptual attributes at various stages of the UNET.
>
> (ii) In this work, we seek to extract such intermediate generalized visual representations from UNET for the quality assessment task. To specifically retrieve quality aware features, we rely upon using cross-attention block outputs, since textual representations can be leveraged to capture quality-specific information from the UNET. As argued by Kumari et. el. [2], cross-attention parameters are more sensitive to finetuning of diffusion models over self-attention and other parameter weights.
>
> (iii) In CLIP-IQA [3], the visual representation is extracted from the final layer of the visual encoder. While the final layer of CLIP model can give coarse perceptual information, we believe that the denoising UNET can capture both fine-grained and coarse visual information due to its multi-scale feature learning architecture. In this work, thus we focus on retrieving multi-scale perceptual information from the cross-attention layer output at all scales.
>
> **Motivation behind using antonym prompts for Quality Estimation:**
>
> (i) While using a single prompt for representing quality, say good quality, images with bad quality can manifest in various ways such as blurriness or compression or ringing artifacts. Such diverse notions of bad quality can make it challenging to align bad quality features together. Thus, specifying bad quality using an initial context such as {'Bad Photo'}, helps more easily align all these different forms of highly distorted images into a unified representation. The same argument also applies if only bad quality is used as a single prompt. The use of antonym prompts makes it easier to unify different distortion types.
> We will add these additional details in our revised manuscript.
>
> **Q2) Similarly, the integration of SlowFast networks with the SD model lacks intuitive discussion (e.g., from the "For TQM..." paragraph onward). The description is very "dry", relying on just providing the equations with minimal explanation of design choices, making it hard to follow. Additionally, Figure 2 is not sufficiently clear or intuitive.**
>
> Ans:  While, Stable diffusion model (SDM) can capture spatial quality attributes in video frames, like in GenzIQA, temporal sub-sampling on videos removes motion-related artefacts such as motion blur, and video stabilization error. For this purpose, we use the SlowFast Network [4] to compensate for the loss in temporal information. The SlowFast network has been pre-trained on the video action recognition dataset viz. Kinetics-400K. The slow pathway in SlowFast network is designed to capture spatial information in video frames from sub-sampled videos, while the fast pathway is designed to capture temporal information from the full-length video. As argued by Feichtenhofer et. al. [4], the fast pathway has better temporal modeling capability when compared to the slow pathway. Thus, we estimate how the similarities of the slow features with the SDM features differ with the similarities of the fast features with the SDM features through a network to compensate for the potential loss due to subsampling.
>   We will include the above intuitive discussion on the use of slow-fast features to capture the loss of temporal information in SDM features due to subsampling.

---

> ### Author Response · Authors · 2025-08-09
> **Response to Reviewer iFTE.**
>
> **Q3) In the Training Details (subsection 4.1), cross-attention maps are extracted using a random timestep between (0, 100]. The rationale for this randomness—rather than a fixed step (e.g., 50 or 100)—is unexplained. Introducing non-deterministic level of noise during inference seems counterintuitive without justification.**
>
> Ans: We thank the reviewer for raising this subtle but important point. During training, we choose random timesteps between (0,100]. In particular, we primarily performed detailed experiments to study in what range, the noise steps need to lie, for effective quality assessment. This is detailed in Fig. 4 in the manuscript. This analysis shows that the (0,100] range is optimal for quality assessment.
>
> During inference, we choose an arbitrary time step in (0,100] and use the same timestep to infer on all images/ video frames. We apologize for not clearly describing this detail in our work. Indeed, we choose a time step of 50 for all our experimental results. Nevertheless, we also analyze the impact of changing this fixed time step chosen in (0,100] in Table 1 below on the FLIVE validation split. We find minimal variation in performance with respect to different choices of the time step in (0,100].
>
> Table1 : SRCC comparison with varying timestep on official FLIVE validation split
>
> | Timestep | SRCC  |
> |----------|-------|
> | 10       | 0.564 |
> | 30       | 0.567 |
> | 50       | 0.569 |
> | 70       | 0.568 |
> | 90       | 0.565 |
>
> We will add these details in our paper.
>
> **Q4) The use of SlowFast networks is ambiguous: it is unclear whether they are pre-trained, how they were pre-trained, or which layers are used for feature extraction.**
>
> Ans: The SlowFast network used in our method is pre-trained on video action recognition dataset viz. Kinetics-400. We extract slow and fast pathway respectively from the pre-final layer of each pathway before the concatenation of the individual pathway's features  of the SlowFast network. We will calrify this detail in our paper and  will release the code for our model upon the acceptance of our manuscript.
>
> References:
>
> [1] Robin Rombach et al., High-resolution
> image synthesis with latent diffusion models. In Proceedings of the IEEE/CVF conference on computer
> vision and pattern recognition, pp. 10684–10695, 2022.
>
> [2] Kumari N et al., Multi-concept customization of text-to-image diffusion. In Proceedings of the IEEE/CVF conference on computer vision and pattern recognition 2023 (pp. 1931-1941).
>
> [3] Jianyi Wang et al., Exploring clip for assessing the look and feel of
> images. In Proceedings of the Thirty-Seventh AAAI Conference on Artificial Intelligence, AAAI’23. AAAI
> Press, 2023a. ISBN 978-1-57735-880-0.
>
> [4] Christoph Feichtenhofer et al., Slowfast networks for video recognition. In Proceedings of the IEEE/CVF international conference on computer vision, pp. 6202–6211,
> 2019.

---

### Review · Reviewer_EQgr · 2025-08-09

**Summary Of Contributions:**

This study investigates the problem of Image and Video Quality Assessment, proposing a novel framework that estimates quality by aligning feature representations derived from a text encoder with those obtained from a diffusion-based image or video generation model. The method optimizes the correspondence between cross-attention matrix representations of the diffusion model across multiple noise levels. Experimental results demonstrate that the proposed approach consistently outperforms competitive baselines across a broad spectrum of benchmark datasets.

**Audience:**

Yes

**Claims And Evidence:**

Yes

**Requested Changes:**

Although I am not an expert in image or video quality assessment, I have several concerns and questions for the authors:

- Building on the weaknesses noted above, the novelty and contribution of this work relative to prior research remain unclear. The primary innovation appears to lie in the use of a diffusion model as opposed to other vision models.
- As previously mentioned, the proposed approach seems to require a textual description at inference time. While this may also apply to all evaluated baselines, the authors should explicitly discuss this point in the paper.
- My main concern / question to the authors is about generalization. It seems to be one of the main claims of the submission, (as reflected in the title ‘Generalized Image and Video Quality Assessment…’), however, it is not clear to me what specific properties render the method generalized. The authors are encouraged to elaborate on this in the manuscript.
- Please clarify and provide further details on the role and importance of textual descriptions within the proposed framework.
- Additional clarification is needed regarding the nature of the ground-truth (GT) scores used for training the model.
- While the paper reports correlation scores between predicted quality and GT human opinion scores, more information is needed about the data collection process: How were the human opinion scores obtained? How many ratings were collected per image? What was the inter-annotator agreement? etc.
- In Table 12, the latency of the proposed method is reported. It would be informative to also include the parameter count for each method.

**Strengths And Weaknesses:**

**Strengths:**
- The proposed method is simple and intuitive to use.
- Experimental results suggest the proposed method is superior to the evaluated baselines across large set of datasets. The authors additionally provide ablation studies to better understand the impact of each component composing their method. Lastly the authors provide runtime analysis of the proposed method compared to the evaluated baselines.

**Weaknesses:**
- Novelty seems a bit limited, mainly proposing to use a diffusion model compared to other image representation models.
- It seems the proposed method requires textual description of the image also during inference time. This is not discussed in the paper.
- It is not clear to me what makes this method ``Generalized''. Which seems to be the main claim of the authors.

---

> ### Author Response · Authors · 2025-08-18
> **Response to Reviewer EQgr**
>
> We thank the reviewer for the detailed comments made about our work. We appreciate that the reviewer finds the detailed experimental analysis of this work useful for understanding of each components of GenzIQA and GenzVQA. We provide a detail answers to the queries raised by the reviewer below.
>
> **Q1) Building on the weaknesses noted above, the novelty and contribution of this work relative to prior research remain unclear. The primary innovation appears to lie in the use of a diffusion model as opposed to other vision models.**
>
> Ans: We agree with the reviewer that diffusion model plays a key role in design of GenzIQA and GenzVQA. However it is not-trivial to apply diffusion models for image/video quality assessment, as we argue below.
>
> (a) Diffusion models are typically designed for generation of different image content conditioned on textual descriptions and their application to the image quality assessment (IQA) task is non-trivial. In particular, arbitrary noise levels cannot help extract quality aware features. In Sec. 4.4.1 (Fig. 3), we see that noise level used to generate images using Stable diffusion model impacts both perceptual quality (LPIPS scores) and the content. Thus, the multi-scale cross-attention blocks of UNet have to be learned at a certain noise level to extract quality and content features of the image/video frame. In Sec. 4.4.2 (Fig. 4) we infer that learning at a noise level of $(0-100]$ gives the best performance in terms of correlation with ground truth human opinion score.
>
> (b) Since diffusion models are trained to exploit the textual prompts for generation, there is a need to understand how textual prompts can be used to extract quality aware information. Thus, we learn the cross-attention block with quality-aware textual embeddings to extract perceptual quality representations through visual query features of the UNet.
>
> (c) Finally, applying text-to-image diffusion models on every video frame is computationally expensive, so we sub-sample the video at 1 frame-per-second. To learn motion-related distortions, which could otherwise got obliterated due to sub-sampling, we propose a novel temporal quality modular (TQM). TQM estimates how the similarities of the visual features given by diffusion model with slow features differs from that of its similarities with the fast features.
>
> As mentioned in the first part of each of the points above, we will add a few lines of clarification regarding why the task of applying diffusion models to the IQA task is non-trivial in the introduction.
>
> **Q2) As previously mentioned, the proposed approach seems to require a textual description at inference time. While this may also apply to all evaluated baselines, the authors should explicitly discuss this point in the paper.**
>
> Ans:     Our methods do not require any image/ video specific textual descriptions either during training or inference time, unlike image generation tasks based on Stable-diffusion. Instead, we use generic quality-aware attributes  such as ('Good Photo.', 'Bad Photo.') during training to capture perceptual representations from the cross-attention blocks for all images/videos as described in Sec 3.2 (contextual prompt tuning). Similarly, during inference, we use the trained embedding using the above context for all the test data. There is no use of content specific text descriptions of images/videos for quality estimation.

---

> ### Author Response · Authors · 2025-08-18
> **Response to Reviewer EQgr**
>
> **Q3) My main concern / question to the authors is about generalization. It seems to be one of the main claims of the submission, (as reflected in the title ‘Generalized Image and Video Quality Assessment…’), however, it is not clear to me what specific properties render the method generalized. The authors are encouraged to elaborate on this in the manuscript.**
>
> Ans: Recently, several works have found that diffusion models achieve superior out of distribution generalization performance over other vision-language models on diverse computer vision tasks such as image retrieval [1], recognition [2], and reasoning [3]. In this work, we leverage this generalization capability of diffusion models for image quality assessment using cross-attention finetuning, quality-aware prompt learning, design of suitable noise levels and temporal distortion information loss estimation for video quality assessment. Thus, our key contribution is to leverage the generalization capabilities effectively for the quality assessment task. In Fig. 6 in the manuscript, we see that across three different datasets, that the GenzIQA features across all scales are separated well for high and low quality images. While in the case of other vision language models such as CLIP-IQA+, there is a significant overlap between the distributions of different image qualities. Thus we are able to leverage diffusion models to achieve generalization across multiple datasets. Our title conveys that we are able to achieve excellent visual quality assessment generalization across different datasets spanning a variety of distortion types and visual content, by leveraging diffusion models. We will add this clarification at the end of the introduction.
>
> **Q4) Please clarify and provide further details on the role and importance of textual descriptions within the proposed framework.**
>
> Ans: In the proposed framework, a fixed pair of textual attributes is used in the overall textual context for learning quality-aware representations in the cross-attention blocks. During inference, the trained textual embedding is used across all test data for quality assessment.
>     Note that no image/ video specific textual descriptions are used either during training or inference, only a generic textual attributes pair is used for guiding/ capturing perceptual-representation.
>
> **Q5) Additional clarification is needed regarding the nature of the ground-truth (GT) scores used for training the model.**
>
> Ans: In quality-assessment literature, the mean quality ratings gathered through human subjective studies, also known as mean opinion score (MOS), is the ground-truth. We use the MOS scores for each image provided with the FLIVE [4] dataset to train GenzIQA. Similarly, we use MOS score of each videos of LSVQ [5] dataset to train GenzVQA. We will add these details in the revised version's training details section.
>
> **Q6) While the paper reports correlation scores between predicted quality and GT human opinion scores, more information is needed about the data collection process: How were the human opinion scores obtained? How many ratings were collected per image? What was the inter-annotator agreement? etc.**
>
> Ans: In this work we have used publicly available IQA and VQA datasets which contain images/videos and their corresponding human-subjective scores, also known as MOS. The subjective studies in these datasets are carried out according to International Telecommunication Union-Telecommunication Standardization Sector (ITU-T) specifications. We note that both the training and testing databases have been published with elaborate studies on the reliability of subjective scores and inter-annotator agreement. In this work we have not carried out any data collection/ subjective evaluation, rather we have used publicly available datasets. More details about data collection done by these datasets can be found in their respective papers, the details of which are given in Appendix A.7 (Tab. 19 and 20).
>
> **Q7) In Table 12, the latency of the proposed method is reported. It would be informative to also include the parameter count for each method.**
>
> Ans: We provide this additional information in the table below and will add it to the respective table in the revised version. Even though the number of parameters of diffusion models are higher than other benchmarking methods, applying the T2I diffusion model at 1 fps keeps the latency during training/ inference within comparable limits.
>
> Table: Number of parameters (in millions) for various Video QA models
>
> | Model        | Parameters (M) |
> |--------------|----------------|
> | VSFA         | 24.1           |
> | CSVT-VQA     | 58.6           |
> | SimpleVQA    | 58.3           |
> | FastVQA      | 28.1           |
> | DOVER        | 56.2           |
> | ModularVQA   | 87.8           |
> | GenzVQA      | 950            |

---

> ### Author Response · Authors · 2025-08-18
> **Response to Reviewer EQgr**
>
> References:
>
> [1] Xuehai He, et al., Discriminative diffusion models as few-shot vision and language
> learners. arXiv preprint arXiv:2305.10722, 2023.
>
> [2] Li, Alexander C., et al. "Your diffusion model is secretly a zero-shot classifier." Proceedings of the IEEE/CVF International Conference on Computer Vision. 2023.
>
> [3] Bahjat Kawar, et al., Imagic: Text-based real image editing with diffusion models. In Proceedings of the IEEE/CVF
> Conference on Computer Vision and Pattern Recognition, pp. 6007–6017, 2023.
>
> [4] Ying, Zhenqiang, et al. "From patches to pictures (PaQ-2-PiQ): Mapping the perceptual space of picture quality." Proceedings of the IEEE/CVF conference on computer vision and pattern recognition. 2020.
>
> [5] Ying, Zhenqiang, et al. "Patch-vq:'patching up'the video quality problem." Proceedings of the IEEE/CVF conference on computer vision and pattern recognition. 2021.

---

> > ### Comment · Reviewer_EQgr · 2025-10-03
> > **Response to the authors**
> >
> > I would like to thank the authors for their detailed response and clarifications.
> >
> > **On the novelty of the proposed approach:** I appreciate the clarification provided. While I understand the authors’ perspective, I believe that points (b) and (c) are not specific to diffusion models but rather apply broadly to text-image models. With respect to noise levels, I agree that the intrinsic nature of diffusion models entails variability in outputs, and the proposed solution of employing multiple noise levels to balance this effect is both straightforward and effective.
> >
> > **On the requirement of textual descriptions:** Thanks for the clarification. I encourage the authors to make this explicit in the main body of the paper.
> >
> > **On generalization claims:** While I acknowledge the authors’ argument that prior work has shown some empirical evidence of diffusion models generalizing to out-of-distribution data and tasks, I find the framing of their method as “generalized” overstated and potentially misleading. The current writeup makes the impression there is an inherent generalization property in the proposed method, while in fact it is limited to leveraging representations from a diffusion model. Moreover, the authors evaluated Stable Diffusion v2 only, hence it is unclear whether the observed behavior reflects diffusion models in general or is it a Stable Diffusion v2 property.
> >
> > **On ground-truth human opinion scores:** Similarly, the current phrasing creates the impression that the authors conducted an original human evaluation, whereas in practice they rely on pre-existing human annotations provided by a benchmark.

---

> > > ### Author Response · Authors · 2025-10-07
> > > **Response to Reviewer EQgr**
> > >
> > > We thank the reviewer for further comments and suggestions.
> > >
> > > 1) We will update the revised version with details about the textual descriptions and ground-truth opinion scores.
> > >
> > > 2) **On generalization claims**: In the quality assessment (QA) literature, our work conducts one of the largest generalization studies across 6 IQA datasets and 11 VQA datasets. Since the diffusion model based representations enable such superior and robust generalization in quality assessment, we believe that it is important to highlight this aspect with respect to quality assessment.

---

### Review · Reviewer_8hmB · 2025-08-23

**Summary Of Contributions:**

Apologies for the late review. I hope the authors can still find the feedback useful for the updated manuscript.

**Summary**:

The authors present methods for No Reference (NR)-QA methods for images and videos. The main idea is to leverage cross-attention representations between image and quality relevant text features,
and prompt tuning to achieve better empirical performance on a variety of
datasets. Specifically, for videos, the authors propose to leverage text-to-image LDMs for extracting representations from sub-sampled videos. However, the drop in motion information is compensated by learning a module that accounts for this loss of motion information. The authors show the performance gains of their method on a number of benchmarks.

**Audience:**

Yes

**Claims And Evidence:**

Yes

**Requested Changes:**

See weaknesses above

**Strengths And Weaknesses:**

**Strengths**:

1. I found the approach quite simple and intuitive. Though I have some general comments about the method (see weaknesses below).
2. The claims in the paper are supported by very rigorous experiments on different benchmarks which establishes the empirical performance of the method.

**Weaknesses**:

1. In the introduction, the authors can maybe explain the setup of No-Reference (NR) QA in more details. In my understanding, NR is a setup where the only information passed to the algorithm is the image for which the quality needs to be assessed.
2. In Section 3.2, the authors propose “We obtain q(x) through a single-step denoising of zt.” Can the authors explain why they use a single denoising step. Im guessing its because of computational reasons as you might need to store multiple denoising calls in memory while training? It would be great if the authors can clarify more on this.
3. In Section 3.2 in Contextual Prompt Tuning, is there a rationale for choosing these prompts? Since the prompts good and bad images are quite subjective (for instance an image could be qualitatively good in different aspects and bad at other aspects at the same time). Hence justifying this rationale is important.
4. Can the authors state the training loss for the IQA and VQA tasks more formally? Its clear from the main text but for I think a more formal equation would improve readability. I also think that Fig. 1 can benefit by highlighting the trainable parameters using a different color. Similarly for Figure 2 for Video QA. Moreover the caption in Figure 2 can be elaborated more so the high level idea is clearer.
5. Can the authors elaborate more on the design on the TQM block as I found it a bit confusing in the main text? From my understanding, the main idea is to attend between the features of the fast and slow networks with the representations in the denoiser to get cross attention maps which are then spatially pooled and concatenated into a representation? Im not sure about the intuition behind calling this final concatenated vector the so-called “temporal-correction factor”. Can the authors elaborate more on this?

---

> ### Author Response · Authors · 2025-09-01
> **Response to Reviewer 8hmB**
>
> We thank the reviewer for the detailed comments made about our work. We appreciate that the reviewer found our method simplistic and also the experimental study to be rigorous. We provide a detail answers to the queries raised by the reviewer below.
>
> Q1) **In the introduction, the authors can maybe explain the setup of No-Reference (NR) QA in more details. In my understanding, NR is a setup where the only information passed to the algorithm is the image for which the quality needs to be assessed.**
>
> Ans: We will update the revised manuscript with more details of the No-reference (NR) setting in
> quality-assessment (QA). In brief, there are three major categories of QA, 1) Full-reference QA: Here the pristine/undistorted version of the inference image/video is available. Examples
> of such measures include as SSIM, and VMAF. 2) Reduced-reference QA: Here some partial
> information about the pristine version of the inference image/video is available. RRED/ST-
> RRED are examples of such methods. 3) No-reference QA: Here only the inference image/video
> is available for quality assessment.
>
> Q2) **In Section 3.2, the authors propose “We obtain q(x) through a single-step denoising
> of zt.” Can the authors explain why they use a single denoising step. Im guessing
> its because of computational reasons as you might need to store multiple denoising
> calls in memory while training? It would be great if the authors can clarify more
> on this.**
>
> Ans: We agree with the reviewer that the primary reason for choosing single-step denoising process in
> the training is the computational overhead of training with multiple-step process. Specifically
> for GenzVQA, even after sub-sampling at 1 fps, the number of frames per-video to be processed
> is between 10 and 30, which will astronomically increase the training overhead for multistep
> denoising. We believe that even with single-step denoising, we are able to extract quality aware
> features with a suitable noise level added to the image. We compare single-step and multi-step
> denoising during inference in Table 13 in the Appendix A.1.
>
> Q3) **In Section 3.2 in Contextual Prompt Tuning, is there a rationale for choosing these
> prompts? Since the prompts good and bad images are quite subjective (for instance
> an image could be qualitatively good in different aspects and bad at other aspects
> at the same time). Hence justifying this rationale is important.**
>
> Ans: ‘Good Photo’ and ‘Bad Photo.’ are initial attributes chosen as input to the CLIP-text encoder
> to provide some grounding to the prompt. Indeed, we add learnable contexts to more care-
> fully prompt the diffusion mode. We see that learning the textual embedding layer with these
> antonym prompts gives significant increment in performance Row 5 of Tab 4 as opposed to
> keeping these prompts fixed in Row 3 of Table 4. Further, in Tables 9, 10 and 11, we explore
> different choices of the fixed prompts and observe little variation in performance with respect
> to this choice, especially when used in conjunction with learnable contexts. Moreover, use of
> such initial attributes gave faster training convergence and also more accurate results rather
> than using only learnable contexts.
>
> Q4) **Can the authors state the training loss for the IQA and VQA tasks more formally?
> Its clear from the main text but for I think a more formal equation would improve
> readability. I also think that Fig. 1 can benefit by highlighting the trainable
> parameters using a different color. Similarly for Figure 2 for Video QA. Moreover
> the caption in Figure 2 can be elaborated more so the high level idea is clearer.**
>
> Ans: We use a mean-squared error loss between the predicted quality scores and the ground truth
> human opinion scores to train both GenzIQA and GenzVQA. Though we have mentioned it in
> end of Sec. 3.2, we will explicitly add equations in the updated manuscript. In Fig.1 and 2, the
> fire and ice symbols denote the parameters for IQA and VQA methods that are trainable. In
> addition, we will elaborate on the caption for Figure 2 in the updated manuscript.
>
> Q5) **Can the authors elaborate more on the design on the TQM block as I found it
> a bit confusing in the main text? From my understanding, the main idea is to
> attend between the features of the fast and slow networks with the representations
> in the denoiser to get cross attention maps which are then spatially pooled and
> concatenated into a representation? Im not sure about the intuition behind calling
> this final concatenated vector the so-called “temporal-correction factor”. Can the
> authors elaborate more on this?**
>
> Ans: The reviewer’s understanding of our design is indeed largely accurate. However, we map this
> representation to a scalar correction factor γp which gets multiplied with the quality score
> qp(vs). The goal of γp is to compensate for the loss of temporal information while estimating
> quality using subsampled frames. γp amplifies/dampens the baseline quality qp(vs) to improve
> the overall prediction.

---

### Decision · Action_Editor_5yY2 · 2025-11-03

**Recommendation:** Accept with minor revision

**Additional Comments:**

Following the authors’ revision, two reviewers lean toward acceptance, acknowledging the paper’s positive contributions, particularly its strong results and comprehensive evaluation. One reviewer (Reviewer EQgr), however, leans toward rejection, citing remaining concerns about overstatements regarding generalization, including: (1) the claimed connection to “generalization” is weak, as the method’s use of diffusion models does not clearly demonstrate inherent generalization properties; and (2) the evaluation relies solely on Stable Diffusion v2, making it unclear whether the findings reflect a general property of diffusion-based models or are specific to this instance. Additional concerns relate to the clarity and completeness of certain descriptions in the modeling and evaluation details.

I agree with Reviewer EQgr that these concerns are valid. However, I believe the contributions of this work remain valuable to the image and video QA community despite these limitations. In particular, the proposed approach demonstrates strong cross-dataset generalizability, outperforming prior methods across diverse datasets and scenarios. The authors have also provided additional explanations and clarifications in their revision.

​​Given these points, I recommend acceptance of this paper. I have the following suggestions for the authors as they prepare the final version:

* Revise the claims throughout the paper, including the title, to be more precise and appropriately scoped, making it clear that the generalization discussed refers specifically to cross-dataset generalization.

* Provide a more detailed and precise explanation of the source of generalization, clarifying whether it primarily arises from the proposed approach itself or from properties of the pretrained diffusion models.

* If possible, include additional results using other types of diffusion models, along with a corresponding discussion, to address Reviewer EQgr’s concern about whether the improvements reflect a general property of diffusion-based models or are specific to Stable Diffusion v2.

* Incorporate all reviewer feedback, including the comments from Reviewer EQgr, in the final revision.

**Audience:**

Yes

**Audience Explanation:**

Researchers and practitioners working on image and video quality assessment, as well as text-to-image or text-to-video generation, may find this paper valuable. In addition, those studying the properties and behaviors of diffusion models may discover some of the paper's findings to be particularly relevant.

**Claims And Evidence:**

Yes

**Claims Explanation:**

Summary:

This paper proposes a diffusion-based framework for no-reference image and video quality assessment (QA) that leverages cross-attention representations from pretrained text-to-image latent diffusion models (LDMs). The key idea is to estimate perceptual quality by aligning quality-aware text prompts with image or video frame representations extracted from intermediate denoising layers. For videos, the method applies LDMs to temporally sub-sampled frames to reduce computational cost, while a temporal quality modulator compensates for the loss of motion information. Cross-dataset experiments demonstrate that the proposed approach achieves superior generalization and state-of-the-art performance across diverse image and video quality benchmarks.

Claims:

The key claims made in the paper are that (1) features extracted from diffusion models offer strong advantages for cross-dataset image and video quality assessment (QA) due to their superior out-of-distribution generalization, as demonstrated in other tasks such as image recognition; (2) extracting quality-aware features from diffusion models for QA is non-trivial, particularly in selecting appropriate noise levels and effectively leveraging contextual text prompts; and (3) the proposed approach achieves state-of-the-art generalizable QA performance across diverse datasets, including user-generated, restoration, variable frame-rate, Ultra-HD, and streaming video content.


Evidence:

The claims are generally supported by the characteristics of the proposed method and the experimental results presented in both the original submission and the revision. However, there are some overstatements regarding the claimed superior generalization, which should be addressed in the final camera-ready version.

---

> ### Author Response · Authors · 2025-11-15
> **Response to Action Editor**
>
> We thank the AE for the valuable comments made about our submitted manuscript. We appreciate that the AE and the reviewers found our work to be acceptable for publication in TMLR. We have uploaded the revised manuscript with all the changes incorporated in it. We provide a detail answers to the changes asked by the AE below.
>
> **Q1) Revise the claims throughout the paper, including the title, to be more precise and appropriately scoped, making it clear that the generalization discussed refers specifically
> to cross-dataset generalization.**
>
> Ans: We have revised the claims in the paper, so that it is clear that we claim generalization with respect to cross-dataset performance. Further, we have revised the title to "Image and Video Quality Assessment using Prompt-Guided Latent Diffusion Models for Cross-Dataset Generalization" in the updated manuscript.
>
> **Q2) Provide a more detailed and precise explanation of the source of generalization, clarifying whether it primarily arises from the proposed approach itself or from properties
> of the pretrained diffusion models.**
>
> Ans: n the updated manuscript’s introduction (fourth paragraph), we provide a precise explanation that
> we leverage the generalization property of diffusion model. Specifically, we leverage the generaliza-
> tion capability of diffusion models for the IQA task by using cross-attention finetuning, quality-aware
> prompt learning, and the design of suitable noise levels to extract quality-aware features from inter-
> mediate layers of the denoiser of the reverse diffusion process.
>
> **Q3) If possible, include additional results using other types of diffusion models, along with
> a corresponding discussion, to address Reviewer EQgr’s concern about whether the
> improvements reflect a general property of diffusion-based models or are specific to
> Stable Diffusion v2.**
>
> Ans: We have included results related to the performance improvement for a different variant of Stable
> diffusion model viz. version 1.5. We have shown in Sec. 4.4.7 (Tab. 7) that across multiple IQA
> datasets, GenzIQA performs considerably better for both Stable diffusion v-1.5 and v-2 base variants.
>
> **Q4) Incorporate all reviewer feedback, including the comments from Reviewer EQgr, in the
> final revision.**
>
> Ans: We have incorporated all reviewer’s feedback including Reviewer EQgr. In our earlier revision, we
> had incorporated all the feedback comments made by the reviewers. In the current revised version,
> we included the latest feedback of Reviewer EQgr in Sec. 3.2 and 4.3.